# A new extended gumbel distribution: Properties and application

**Aisha Fayomi**[1☯], **Sadaf Khan** [2☯]*, **Muhammad Hussain Tahir**[2☯], **Ali Algarni**[1☯], **Farrukh Jamal**[2☯], **Reman Abu-Shanab**[3☯]

**1** Faculty of Science, Department of Statistics, King Abdulaziz University, Jeddah, Saudi Arabia, **2** Department of Statistics, The Islamia University of Bahawalpur, Bahawalpur, Pakistan, **3** Department of Mathematics, College of Science, University of Bahrain, Zallaq, Bahrain

☯ These authors contributed equally to this work.
* smkhan6022@gmail.com

**Data Availability Statement:** All relevant data are within the paper.

**Funding:** The author(s) received no specific funding for this work.

## Abstract

A robust generalisation of the Gumbel distribution is proposed in this article. This family of distributions is based on the T-X paradigm. From a list of special distributions that have evolved as a result of this family, three separate models are also mentioned in this article. A linear combination of generalised exponential distributions can be used to characterise the density of a new family, which is critical in assessing some of the family's properties. The statistical features of this family are determined, including exact formulations for the quantile function, ordinary and incomplete moments, generating function, and order statistics. The model parameters are estimated using the maximum likelihood method. Further, one of the unique models has been systematically studied. Along with conventional skewness measures, MacGillivray skewness is also used to quantify the skewness measure. The new probability distribution also enables us to determine certain critical risk indicators, both numerically and graphically. We use a simulated assessment of the suggested distribution, as well as apply three real-world data sets in modelling the proposed model, in order to ensure its authenticity and superiority.

## 1 Introduction

The employment of traditional probability models to anticipate real-life occurrences is causing increasing dissatisfaction among applied practitioners. Tail characteristics and goodness of fit metrics may have a constraining tendency, which could be one of the reasons. As a response, in recent years, there has been a substantial rise in the generalisation of well-known probability distributions. The challenge is to find such versatile families that can fit both skew and symmetric data. It's important to realize that the majority of generalised distributions described in the literature are constructed using the generalised classes approach (G-class) and the compounding principle. In [1], the authors provided a concise review of literature regarding generalization of distributions and transformation through versatile parameter induction techniques. We offer to the readers a few, but not exhaustive, lists in chronological sequence: [2–14].

**Competing interests:** The authors have declared that no competing interests exist.

According to [15], Emil J. Gumbel originated the use of the Gumbel distribution (GuD) on data bearing extreme values. By "extreme data" we mean the behaviour of a random variable that occurs at the sample threshold level and is seen using intense data and insights. In the reference [16], the authors defined the GuD, alternatively known as extreme value distribution-type I ($\gamma = 0$), as the predominant model for quantifying extreme occurrences such as flood frequency analysis, droughts, earthquakes, sea currents and wind speed in order to understand the trajectory, magnitude, and pattern of complex phenomena. Environmental sciences, geology, accelerated life testing, meteorology, risk assessment and epidemiology are just a few of the a set of diverse fields where it might well be utilized. The authors in [17] showed that the score statistics of global sequence alignment follows a Gumbel distribution. In the reference [18], a comprehensive list of real life scenarios to which GuD can be applicable is provided by the authors. To learn more, see references [19–24].

Earliest generalizations of GuD was reported by [25] by introducing a shape parameter to Gumbel distribution. [26] provided a generalization of GuD based on the asymptotic distribution of the $m$th extreme, tracing back to [27]. [28] emphasized on a trivial choice of distribution since the GuD with only location and scale parameter yields narrower confidence intervals and has also the danger of underestimating the level of return. In the reference [29], a unique modification of GuD was proposed. It is based on the logarithmic transformation of an odd Weibull variable and is defined as

$$F(x; \beta, \mu, \sigma) = 1 - \left[1 + (e^{\Delta} - 1)^{\beta}\right]^{-1}, \tag{1}$$

where $\Delta = e^{(x-\mu)/\sigma}$, $-\infty \leq x \leq +\infty$, $-\infty \leq \mu \leq +\infty$ and $0 \leq \beta\sigma \leq +\infty$.

Since then, researchers adopted a more formalistic approach to generalize GuD. Some notable generalizations include [30] to define Beta-Gumbel (BGu), [31] to propose Kumaraswamy-gumbel (KumGu), [32] to define exponentiated-Gumbel (EGu), [33] to define exponentiated-generalized Gumbel (EGGu), just to mention some.

[14] proposed a simplified approach to generalize any continuous distribution viz. a viz. the *transformed-transformer (T-X)* family, which has become an indispensable part of modern distribution theory. Let $z(t)$ be the probability density function (pdf) and $Z(t)$ be the cumulative distribution function (cdf) of a random variable (av) $T$ such that $t\epsilon(a_1, a_2)$ with support $-\infty < a_1 < a_2 < \infty$. Let $W[Z(x)]$ act as generator function of the cdf $Z(x)$ of any baseline av such that $K[Z(x)]$ is differentiable and increasing, lies in the defined range, i.e. $a_1 \leq K[Z(x)] \leq a_2$ such that when $x \rightarrow -\infty$ as $K[Z(x)] \rightarrow a_1$ and $x \rightarrow +\infty$ as $K[Z(x)] \rightarrow a_2$.

$$F_{TX}(x; \varphi) = \int_{a_1}^{K[Z(x;\varphi)]} z(t)\, dt = Z(K[Z(x;\varphi)]). \tag{2}$$

The pdf corresponding to Eq (2) is

$$f_{TX}(x; \varphi) = z\left(K[Z(x;\varphi)]\right) \frac{d}{dx} K[Z(x;\varphi)]. \tag{3}$$

To generalize any continuous distribution, the methodology defined by the cdf in Eq (2) has become indispensable part of modern distribution theory. In the same vein, Al-Aqtash et al. [34] introduced the Gumbel-X family of distributions. Let $z(t)$ be the density function (pdf) and $Z(t)$ be the distribution function (cdf) of an arbitrary variable (av) $T$ such that $t\epsilon(a_1, a_2)$ with support $-\infty < a_1 < a_2 < \infty$. Let $K[Z(x; \varphi)] = \log\left[(Z(x; \varphi))/1-(Z(x; \varphi))\right]$ act as the generator function of the cdf of any baseline av such that $K[Z(x; \varphi)]$ fulfills the defined

criterion. For $\mu = 0$, the cdf of Gumbel-X family is given as

$$F(x; \sigma) = \mathrm{e}^{-\Delta^{-1/\sigma}} \qquad x \, \epsilon \, \mathbb{R}, \, \sigma > 0, \tag{4}$$

where $\Delta = Z(x)/(1 - Z(x))$.

This study introduces a new class of distributions following the T-X methodology, viz. a viz. *the exponentiated Gumbel-G (EGuG) family of distributions*. This is achieved by replacing the link function $K[Z(x; \varphi)] = \log[-\log(1 - Z(x; \varphi))]$ in T-X family. It is worthy to remember that the link function $\log[-\log(1 - Z(x; \varphi))] = \log[-\log(Z(x; \varphi))]$ and either of the mentioned form can be employ. EGuG family has thus far not been reported in the literature. We choose EGu distribution to define new family due to its superiority over the ordinary GuD because of presence of shape parameter $\theta$ that entails the improvements in tail of the distribution. Moreover, to the best of our knowledge, majority of the extreme value theory literature is supported by data from meteorology, geology, seismology, and hydrology (see references [15, 18–22, 24, 30–34]). The health implications of climate-related shifts in extreme event exposure, on the other hand, have not been explored. This study's theoretical investigation will presumably fill this void in existing literature. Additionally, $\log[-\log(1 - Z(x; \varphi))]$ function involves double log transformation and cannot be employed on GuD, which somehow makes the link function redundant. Following the success of the proposed generator to generalize Logistic and Normal distributions, we use this generator to define EGuG distribution. We study some of its mathematical properties and provide general properties and application of one of its specific model.

This article is outlined as follows: In Section 2, we define the EGuG family and present some of its special models. In order to optimise the structure of the generalisation being proposed, we provide the linear representation of EGuG density along with some of the mathematical properties of the family such as shapes of density and hazard rate function, moments and generating function, order statistics and estimation of model parameters. In Section 3, we choose Nadarajah-Haghighi (NH) distribution as baseline model to form EGuNH distribution whose mathematical properties as well as some risk measures are established. A simulation study is also conducted for some parametric combinations. Section 4 comprises of the numerical illustrations based on three life data sets. In Section 5, the article's concluding thoughts are summed up.

## 2 The EGuG family

Let $T$ follows the EGu av with $\mu = 0$ and shape parameters $\theta \geq 0$ and $\sigma \geq 0$, say EGu $(\theta, \sigma)$, then its cdf is given by

$$Z(t; \varphi) = 1 - (1 - \mathrm{e}^{-\mathrm{e}^{-t/\sigma}})^{\theta}, \, t \, \epsilon \, \mathbb{R}. \tag{5}$$

The corresponding pdf to Eq (5) is given as

$$z(t; \varphi) = \frac{\theta}{\sigma} (1 - \mathrm{e}^{-\mathrm{e}^{-t/\sigma}})^{\theta-1} \, \mathrm{e}^{-\mathrm{e}^{-t/\sigma}} \, \mathrm{e}^{-t/\sigma}. \tag{6}$$

For any baseline distribution with cdf $Z(x; \Phi)$ and pdf $z(x; \Phi) = dZ(x; \Phi)/dx$, the cdf of EGuG family is given as

$$
\begin{aligned}
F(x; \theta, \sigma, \varphi) &= 1 - \int_{a_1}^{K[Z(x;\varphi)]} z(t) \, dt \\
&= [1 - \exp\{-(-\log\{Z(x; \Phi)\})^{-1/\sigma}\}]^{\theta}, \quad x > 0, \, \theta, \sigma > 0.
\end{aligned}
\tag{7}
$$

where $\theta, \sigma$ are shape parameters and $\varphi$ is the vector of baseline parameter.

The pdf corresponding to Eq (7) is given by

$$
\begin{aligned}
f(x; \theta, \sigma, \varphi) &= \frac{\theta\, z(t; \varphi)}{\sigma\, Z(t; \varphi)} [-\log\{Z(t; \varphi)\}]^{-(1/\sigma)-1} \exp\{-[-\log\{Z(t; \varphi)\}]^{-1/\sigma}\} \\
&\quad \times [1 - \exp\{-[-\log\{Z(t; \varphi)\}]^{-1/\sigma}\}]^{\theta-1},
\end{aligned}
\tag{8}
$$

where $Z(x; \varphi)$ is the baseline cdf and $z(x; \varphi)$ is the baseline pdf. Furthermore, the dependence on the vector $\varphi$ of the parameters might be omitted at times and simply write $Z(x) = Z(x; \varphi)$ and $z(x) = z(x; \varphi)$. Henceforth, $X \sim EGu{-}G(\theta, \sigma; \varphi)$ denotes an av having density Eq (8).

The survival function (sf), hazard rate function (hrf) and cumulative hazard rate function (chrf) of this new family are, respectively, given by

$$
S(x) = 1 - [1 - \exp\{-(-\log\{Z(x)\})^{-1/\sigma}\}]^{\theta},
$$

$$
h(x) = \frac{\theta\, z(x) \exp\{-[-\log\{Z(x)\}]^{-1/\sigma}\}[1 - \exp\{-[-\log\{Z(x)\}]^{-1/\sigma}\}]^{\theta-1}}{\sigma\, Z(x)\, [-\log\{Z(x)\}]^{\frac{1}{\sigma}+1}[1 - \{1 - \exp(-[-\log\{Z(x)\}]^{-1/\sigma})\}^{\theta}]}
$$

and

$$
H(x) = -\ln[1 - \{1 - \exp(-[-\log\{Z(x)\}]^{-1/\sigma})\}^{\theta}].
$$

Simulating the EGuG family is simply done by inverting Eq (7) as follows: If $U$ has a uniform $U(0, 1)$ distribution, then

$$
x = Q_Z\left(e^{-[\log(1-u^{1/\alpha})^{-1}]^{-\sigma}}\right)
\tag{9}
$$

has the density function Eq (8), where $Q_Z(.) = Z^{-1}(.)$ is the baseline quantile function (qf).

## 2.1 Special models

Eq (7) can be useful in modelling real life survival data with different shapes of hrf. Table (1) lists $-\log[Z(x; \varphi)]$ and the associated parameters for some special distributions.

Here three special models of EGuG family of distribution are defined.

**2.1.1 EGu-Weibull(EGuW).** The EGu-W model is defined from Eq (7) by taking $Z(x; \varphi) = 1 - \exp\{-ax^b\}$, $z(x; \varphi) = abx^{b-1}\exp\{-ax^b\}$, as cdf and pdf of the baseline Weibull distribution with $a, b > 0$, respectively.

**Table 1. Distributions and corresponding $-\log[Z(x; \varphi)]$ functions.**

| Distribution | $-\log[Z(x; \varphi)]$ | $\varphi$ |
|---|---|---|
| Burr XII ($x > 0$) | $-\log[1 - (1 + x^a)^{-b}]$ | $(a, b)$ |
| Weibull ($x > 0$) | $-\log[1 - e^{-a x^b}]$ | $(a, b)$ |
| Normal ($-\infty < x < \infty$) | $-\log\left[\phi\left(\frac{x-\mu}{\sigma}\right)\right]$ | $(\mu, \sigma)$ |
| Nadarajah Haghighi ($x > 0$) | $-\log[1 - e^{1-(1+\lambda x)^\alpha}]$ | $(\alpha, \lambda)$ |
| Rayleigh ($x > 0$) | $-\log[1 - e^{-a x^2}]$ | $(a)$ |
| Exponential ($x > 0$) | $-\log[1 - e^{-\alpha x}]$ | $(\alpha)$ |
| Power function ($0 < x < a$) | $-b\log\left[\frac{x}{a}\right]$ | $(a, b)$ |
| Fréchet ($x > 0$) | $\left(\frac{\lambda}{x}\right)^\sigma$ | $(\lambda, \sigma)$ |
| Inverted Rayleigh ($x > 0$) | $\left(\frac{\lambda}{x^2}\right)^\sigma$ | $(\lambda, \sigma)$ |
| Burr III ($x > 0$) | $z\log[1 + x^{-c}]$ | $(c, z)$ |
| Pareto ($\delta < x < \infty$) | $-\log[1 - (\delta/x)^\lambda]$ | $(\delta, \lambda)$ |

The cdf and pdf of EGu-W distribution are, respectively, given by

$$F(x; \theta, \sigma, a, b) = [1 - \exp(-\{-\log(1 - e^{-a\,x^b})\}^{-1/\sigma})]^{\theta}, \ x > 0 \quad \theta, \sigma, a, b > 0,$$

and

$$
\begin{aligned}
f(x; \theta, \sigma, a, b) \ &= \ \frac{\theta\, a\, b\, x^{b-1}\, e^{-a\,x^b}}{\sigma[1 - e^{-a\,x^b}]}[-\log(1 - e^{-a\,x^b})]^{-(1/\sigma)-1} e^{-[-\log(1-e^{-a\,x^b})]^{-(1/\sigma)}} \\
&\quad \times [1 - \exp(-\{-\log(1 - e^{-a\,x^b})\}^{-1/\sigma})]^{\theta-1},
\end{aligned}
$$

where $\theta$, $\sigma$ and $b$ are shape parameters while $a$ is scale parameter.

**2.1.2 EGu-BurrXII(EGuBXII).**   Let us consider the parent distribution as BXII with power parameters $a, b > 0$ by taking $Z(x; \varphi) = 1 - (1 + x^a)^{-b}$, $z(x; \varphi) = abx^{a-1}(1 + x^a)^{-b-1}$ be the cdf and pdf of the distribution.

The cdf and pdf of EGu-BXII distribution are, respectively, given by

$$F(x; \theta, \sigma, a, b) = [1 - e^{-\{-\log[1-(1+x^a)^{-b}]\}^{-1/\sigma}}]^{\theta}, \ x > 0, \quad \theta, \sigma, a, b > 0$$

and

$$
\begin{aligned}
f(x; \theta, \sigma, a, b) \ &= \ \frac{\theta\, a\, b\, x^{a-1}(1 + x^a)^{-b-1}}{\sigma\{1 - (1 + x^a)^{-b}\}}[-\log\{1 - (1 + x^a)^{-b}\}]^{-(1/\sigma)-1} \\
&\quad \times \exp[-\{-\log(1 - (1 + x^a)^{-b})\}^{-(1/\sigma)}] \\
&\quad \times [1 - \exp(-\{-\log(1 - (1 + x^a)^{-b})\}^{-1/\sigma})]^{\theta-1},
\end{aligned}
\tag{10}
$$

where $\theta$, $\sigma$, $a$ and $b$ are shape parameters.

**2.1.3 EGu-Nadarajah Haghighi(EGuNH).**   Consider to take Nadaraah Haghighi (NH) as baseline distribution with cdf as $Z(x; \varphi) = [1 - \exp\{1 - (1 + \lambda x)^{\beta}\}]$ and pdf as $z(x; \varphi) = \lambda\beta(1 + \lambda x)^{\beta-1}\exp\{1 - (1 + \lambda x)^{\beta}\}$. Then, the cdf and pdf of EGuNH reduces to

$$F(x; \theta, \sigma, \alpha, \lambda) = [1 - \exp\{-(-\log[1 - e^{1-(1+\lambda x)^{\alpha}}])^{-1/\sigma}\}]^{\theta}, \quad x > 0, \quad \theta, \sigma, \alpha, \lambda > 0 \tag{11}$$

and

$$
\begin{aligned}
f(x; \theta, \sigma, \alpha, \lambda) \ &= \ \frac{\theta\alpha\lambda(1 + \lambda x)^{\alpha-1}e^{1-(1+\lambda x)^{\alpha}}}{\sigma[1 - e^{1-(1+\lambda x)^{\alpha}}]}[-\log\{1 - e^{1-(1+\lambda x)^{\alpha}}\}]^{-(1/\sigma)-1} \\
&\quad \times \exp[-\{-\log(1 - e^{1-(1+\lambda x)^{\alpha}})\}^{-1/\sigma}] \\
&\quad \times [1 - \exp\{-(-\log\{1 - e^{1-(1+\lambda x)^{\alpha}}\})^{-1/\sigma}\}]^{\theta-1}.
\end{aligned}
\tag{12}
$$

where $\theta$, $\sigma$ and $\alpha$ are shape parameters while $\lambda$ is scale parameter.

## 2.2 Useful expansion for the EGuG cdf

We provide a useful expansion for Eq (7) in terms of linear combinations of exp-G distribution. For a random baseline cdf $Z(x)$, an av is said to have the exp-G distribution having parameter $\ell$ such that $\ell > 0$, say $Y \sim$ exp-G $(\ell)$, if its pdf and cdf are given as

$$h_{\ell}(x) = \ell\, Z^{\ell-1}(x; \Phi)\, z(x; \Phi) \qquad \text{and} \qquad H_{\ell}(x) = Z(x)^{\ell}$$

respectively. Thus, several properties of the proposed model can be derived from those properties of the exp-G distribution studied by the authors in [3–8], to mention few.

By expanding Eq (7) using binomial and power series expansion, the resultant expression is given

$$F(x) = \sum_{j=0}^{\infty}\sum_{i=0}^{\infty}\binom{\theta}{j}\frac{(-1)^{i+j}j^i}{i!}[-\log Z(x;\Phi)]^{-i/\sigma}. \tag{13}$$

Using `Mathematica` software, it can be verified that we can start the limit of integers $(i, j)$ from 1 instead of 0 in above equation. Further, we can write $[-\log Z(x;\varphi)]^{-i/\sigma}$ as $[-\log\{1 - \bar{Z}(x;\phi)\}]^{-i/\sigma}$ since $Z(x) = 1 - \bar{Z}(x)$.

Now consider, for any real parameter $c$ and $z\epsilon(0, 1)$, the following formula holds:

$$[-\log(1-z)]^\ell = \sum_{k=0}^{\infty}P_k(c)z^{c(m+1)}, \tag{14}$$

where $P_0(c) = 1/2$; $P_1(c) = c(3c + 5)/24$; $P_2(c) = c(c^2 + 5c + 6)/48$ etc is the stirling's polynomial. Then, the cdf $F(x)$ in Eq (13) can be expressed (using Eq (14) and generalized binomial expansion) as

$$F(x) = \sum_{m=1}^{\infty}\Pi_m H_m(x), \tag{15}$$

where $\Pi_m = (-1)^{m+1}\sum_{i,j=1}^{\infty}\sum_{k=0}^{\infty}\frac{(-1)^{i+j}j^i}{i!\,j!}\frac{\Gamma(-i/\sigma(k+1)+m)}{m\,\Gamma(-i/\sigma(k+1))}\binom{\theta}{\sigma}P_k(-i/\sigma)$.

By differentiating Eq (15), we obtain

$$f(x : \sigma, \theta, \Phi) = \sum_{m=1}^{\infty}\omega_m h_m(x), \tag{16}$$

where $h_m(x) = mZ^{m-1}(x;\Phi)z(x;\Phi)$ is the exp-G density function with power parameter $m$.

## 2.3 Shapes of density and hazard function

Analytical descriptions of density and hrf forms are conceivable. The roots of the equation represent the EGuG density's critical points:

$$\frac{z'(x;\Phi)}{z(x;\Phi)} - \frac{z(x;\Phi)z'(x;\Phi)}{Z(x;\Phi)} + \left\{\frac{[-\log Z(x;\Phi)]^{-(1/\sigma)-1}z(x;\Phi)z'(x;\Phi)}{\sigma Z(x;\Phi)}\right\}$$
$$+ \left\{\frac{(\theta-1)[-\log Z(x;\Phi)]^{-(1/\sigma)-1}e^{-[-\log Z(x;\Phi)]^{-1/\sigma}}z(x;\Phi)z'(x;\Phi)}{\sigma Z(x;\Phi)}\right\} = 0. \tag{17}$$

The equation is used to find the EGuG hrf's crucial points.

$$\frac{z'(x;\Phi)}{z(x;\Phi)} + \frac{(\sigma+1)z(x;\Phi)z'(x;\Phi)}{\sigma[-\log Z(x;\Phi)]Z(x;\Phi)} - \left\{\frac{[-\log Z(x;\Phi)]^{-(1/\sigma)-1}z(x;\Phi)z'(x;\Phi)}{\sigma Z(x;\Phi)}\right\}$$
$$+ \left\{\frac{(\theta-1)e^{-[-\log Z(x;\Phi)]^{-1/\sigma}}[-\log Z(x;\Phi)]^{-(1/\sigma)-1}z(x;\Phi)z'(x;\Phi)}{\sigma Z(x;\Phi)[1 - \exp\{-[-\log Z(x;\Phi)]^{-1/\sigma}\}]}\right\}$$
$$+ \left\{\frac{\theta\left[1 - e^{-[-\log Z(x;\Phi)]^{-\frac{1}{\sigma}}}\right]^{\theta-1}e^{-[-\log Z(x;\Phi)]^{-\frac{1}{\sigma}}}[-\log Z(x;\Phi)]^{-\frac{1}{\sigma}-1}z(x;\Phi)z'(x;\Phi)}{\sigma Z(x;\Phi)\left[1 - \{1 - e^{-[-\log Z(x;\Phi)]^{-\frac{1}{\sigma}}}\}^{\theta}\right]}\right\} = 0. \tag{18}$$

Any numerical software can be used to examine Eqs (17) and (18) to determine the local maximum and minimum and inflexion points.

## 2.4 Moments

The first formula for the $s$th moment of $X$ follows from Eq (16) as

$$\mu'_s = \sum_{m=1}^{\infty} \Pi_m E(X_m^s). \tag{19}$$

where $\mathbb{E}(X_m^s) = \int_0^{\infty} x^s h_m(x) dx$. Setting $s = 1$ in Eq (refrthmoment1pdfmix1) can provide explicit expression for the mean of several parent distributions.

A second alternative formula for $\mu'_n$ is obtained from Eq (19) in terms of the baseline qf as

$$\mu'_n = \sum_{i,j=0}^{\infty} \ell\, \omega_{i,j}\, \tau(n, \ell - 1). \tag{20}$$

where $\tau(n, \ell - 1) = \int_0^1 Q_Z(u)^n u^\ell du$.

The central moments $(\mu_t)$ and cumulants $(\kappa_t)$ of $X$ can follow from Eq (19) as $\mu_s = \sum_{k=0}^p \binom{s}{k} (-1)^k \mu_1'^s \mu'_{s-k}$ and $\kappa_s = \mu'_s - \sum_{k=1}^{s-1} \binom{s-1}{k-1} \kappa_k \mu'_{s-k}$, respectively, where $\kappa_1 = \mu'_1$. The skewness $\gamma_1 = \kappa_3 / \kappa_2^{3/2}$ and kurtosis $\gamma_2 = \kappa_4 / \kappa_2^2$ can be calculated from the third and fourth standardized cumulants.

The $s$th incomplete moment of $X$ can be determined from Eq (16) as

$$m_s(y) = \sum_{m=0}^{\infty} m\, \omega_m \int_0^{Z(y)} Q_Z(u)^n u^m d_u. \tag{21}$$

The last integral can be computed for most G distributions.

A crucial applicability of the first incomplete moment $m_1(\cdot)$ has to do with the Bonferroni and Lorenz curves, which are extremely beneficial in a variety of fields. For a given probability $\pi$, they are given by $B(\pi) = m_1(q)/(\pi \mu'_1)$ and $L(\pi) = m_1(y)/\mu'_1$, respectively, where $m_1(y)$ comes from Eq (refincompletepdfmix1) with $s = 1$ and $q = Q(\pi)$ follows from Eq (9). The Lorenz and Bonferroni curve for EGuNH are displayed graphically (Figs 13 and 14, subsequently).

The totality of excursions from the mean and median is used to estimate the degree of scatter in a population and is defined by $\delta_1 = \int_0^{\infty} |x - \mu| f(x) dx$ and $\delta_2(x) = \int_0^{\infty} |x - M| f(x) dx$, respectively, where $\mu'_1 = \mathbb{E}(X)$ is the mean and $M = Q(0.5)$ is the median. These measures can be expressed as $\delta_1 = 2\mu'_1 F(\mu'_1) - 2m_1(\mu'_1)$ and $\delta_2 = \mu'_1 - 2m_1(M)$, where $F(\mu'_1)$ is given by Eq (refcdfEGuG).

The moment generating function (mgf) of $X$ can be expressed as

$$M_X(t) = \sum_{m=0}^{\infty} \omega_m M_m(t),$$

where $M_m(t)$ is the mgf of $Y_m$. Hence, $M(t)$ can be determined from the exp-G generating functions.

## 2.5 Order statistics

Order statistics are used in a wide range of statistical theory and practise. Let $X_1, \ldots, X_n$ is a random sample from the EGu-G distribution and $X_{i:\, n}$ denote the $i$th order statistic. Then, pdf

of $X_{i:n}$ can be written as

$$
\begin{aligned}
f_{i:n}(x) &= \frac{1}{\beta(i, n-i+1)} f(x) F(x)^{i-1} \{1 - F(x)\}^{n-i} \\
&= \frac{1}{\beta(i, n-i+1)} \sum_{j=0}^{n-i} (-1)^j \binom{n-i}{j} f(x) F(x)^{j+i-1}.
\end{aligned}
$$

Inserting Eq (refcdfEGuG) and Eq (refpdfEGuG) in the last equation, and expanding it as in section (3.1), we get

$$
f_{i:n}(x) = \sum_{j=0}^{n-i} \eta_j \, h_m(x), \tag{22}
$$

where

$$
\eta_j = \frac{(-1)^j}{\beta(i, n-i+1)} \binom{n-i}{j} \sum_{m=0}^{\infty} \Pi_m^*
$$

and

$$
\Pi_m^* = (-1)^m \sum_{i,j=1}^{\infty} \sum_{k=0}^{\infty} \frac{(-1)^{i+j} j^i \Gamma\{-i/\sigma(k+1)+m\}}{i! j! \Gamma\{-i/\sigma(k+1)+m\} m} \binom{\theta(i+j)}{j} P_k(-i/\sigma).
$$

## 2.6 Estimation

The three alternate approaches for inference are point estimation, interval estimation, and hypothesis tests. Several approaches for parameter point estimation have been published in the literature, the most extensively utilised of which is the maximum likelihood method. MLEs (maximum likelihood estimates) have properties that can be used to construct confidence ranges for model parameters. Large sample theory provides simple approximations that work well in repeated sampling for these estimations. The normal approximation for MLEs can be tackled analytically or computationally in distribution theory.

We use the optimum likelihood method to estimate the unknown parameters of the new distribution. Let $x_1, \cdots, x_n$ be $n$ observations from the EGu-G family given by Eq (8) with parameter vector $\Theta = (\theta, \beta; \Phi)^\top$. The log-likelihood $\Im = \Im(\Theta)$ for $\Theta$ is given by

$$
\begin{aligned}
&n \log(\theta) - n \log(\sigma) + \sum_{i=1}^{n} \log z(x_i; \Phi) - \sum_{i=1}^{n} \log Z(x_i; \Phi) - \left(\frac{1+\sigma}{\sigma}\right) \sum_{i=1}^{n} \log\left[-\log Z(x_i; \Phi)\right] \\
&- \sum_{i=1}^{n} \left[-\log Z(x_i; \Phi)\right]^{-1/\sigma} + (\theta - 1) \sum_{i=1}^{n} \log\left[1 - \exp\left\{-[-\log Z(x_i; \Phi)]^{-1/\sigma}\right\}\right].
\end{aligned} \tag{23}
$$

Eq (refmleegug) can be maximized either directly by using the R (`optim function`), SAS (`NLMixed` procedure) or Ox (`MaxBFGS` function), or then by solving the nonlinear

likelihood equations by differentiating it. The components of the score vector $U(\Theta)$ are

$$U_\theta = \frac{n}{\theta} + \sum_{i=1}^{n} \log\left[1 - \exp\left\{-[-\log Z(x_i; \Phi)]^{-1/\sigma}\right\}\right],$$

$$U_\sigma = -\frac{n}{\sigma} + \frac{1}{\sigma^2}\sum_{i=1}^{n}\log\left[-\log Z(x_i; \Phi)\right] - \sum_{i=1}^{n}[-\log Z(x_i; \Phi)]^{-1/\sigma},$$

$$U_{\Phi_k} = \sum_{i=1}^{n}\left[\frac{z'(x_i; \Phi)}{z(x_i; \Phi)}\right] - \sum_{i=1}^{n}\frac{z'(x_i; \Phi) z(x_i; \Phi)}{Z(x_i; \Phi)} + \left(\frac{1 + \sigma}{\sigma}\right)\frac{z'(x_i; \Phi) z(x_i; \Phi)}{Z(x_i; \Phi)[-\log Z(x_i; \Phi)]}$$

$$- \sum_{i=1}^{n}\frac{[-\log Z(x_i; \Phi)]^{(-1/\sigma)-1} z(x_i; \Phi) z'(x_i; \Phi)}{\sigma Z(x_i; \Phi)}$$

$$+ (\theta - 1)\sum_{i=1}^{n}\frac{\exp\left[-\{-\log Z(x_i; \Phi)\}\right]\left[-\{-\log Z(x_i; \Phi)\}\right]^{(-1/\sigma)-1} z(x_i; \Phi) z'(x_i; \Phi)}{\sigma Z(x_i; \Phi)[1 - \exp\{-[-\log Z(x_i; \Phi)]^{-1/\sigma}\}]}.$$

Setting these equations to zero and solving them simultaneously yields the MLEs $\hat{\Theta}$ of the family parameters.

The observed information matrix for the parameter vector $\Theta = (\theta, \sigma, \Phi_k)^\top$ is given by

$$J(\theta) = -\frac{\partial^2 \ell(\Theta)}{\partial \Theta \, \partial \Theta^\top} = -\begin{pmatrix} J_{\theta\theta} & J_{\theta\sigma} & J_{\theta\Phi_k} \\ \cdot & J_{\sigma\sigma} & J_{\sigma\Phi_k} \\ \cdot & \cdot & J_{\Phi_k\Phi_\ell} \end{pmatrix},$$

whose elements can be determined by using any mathematical software. Under normal conditions of regularity, the multivariate normal $N_3(0, J(\hat{\Theta})^{-1})$ distribution, where $J(\hat{\Theta})^{-1}$ is the observed information analysed at $\hat{\Theta}$, can be used to estimate confidence ranges for model parameters. Furthermore, we may use likelihood ratio (LR) statistics to assess the EGuG model to any of its specific models.

## 3 Properties of EGuNH

In comparison to Gamma, Weibull, and exponentiated exponential distributions, NH distribution (also known as extended exponential distribution) is the preferred option for zero inflated data. The cdf and pdf of NH distribution has already been defined in Section (3.1.3). For $\lambda = 1$, we define the cdf and pdf of EGuNH distribution as

$$F(x) = [1 - \exp\{-(-\log[1 - e^C])^{-1/\sigma}\}]^\theta, \tag{24}$$

and

$$f(x) = \frac{\theta\alpha(1 + x)^{\alpha-1}e^C}{\sigma[1 - e^C]}[-\log\{1 - e^C\}]^{-(1/\sigma)-1} e^{-\{-\log(1-e^C)\}^{-1/\sigma}}$$

$$\times [1 - \exp\{-(-\log\{1 - e^C\})^{-1/\sigma}\}]^{\theta-1}, \tag{25}$$

where $C = 1 - (1 + x)^\alpha$ and $\theta$, $\sigma$ and $\alpha$ are shape parameters.

Henceforth, we denote by $X$ a av having density (25). The sf and hrf of $X$ has the form

$$s(x) = 1 - [1 - \exp\{-(-\log[1 - e^C])^{-1/\sigma}\}]^\theta$$

and

$$
\begin{aligned}
h(x) \;=\; & \alpha\,\theta\,\sigma^{-1}(x+1)^{\alpha-1}\mathrm{e}^{-C}\,(1-\mathrm{e}^{C})^{-1}[-\log\,(1-\mathrm{e}^{C})]^{-(1/\sigma)-1}\\
& \times[\mathrm{e}^{-[-\log\,(1-\mathrm{e}^{C})]^{-1/\sigma}}]\,[1-\mathrm{e}^{-\{-\log\,(1-\mathrm{e}^{C})\}^{-1/\sigma}}]^{\theta-1}\\
& \times[1-(1-\mathrm{e}^{-\{-\log\,(1-\mathrm{e}^{C})\}^{-1/\sigma}})^{\theta}]^{-1}.
\end{aligned}
$$

## 3.1 Shapes of density and hazard rate function of EGuNH distribution

The crucial points of the pdf of $X$ are obtained from the equation:

$$
\begin{aligned}
& \alpha^{-1}\mathrm{e}^{-C}(x+1)^{1-\alpha}[(\alpha-1)\alpha\,\mathrm{e}^{C}(x+1)^{\alpha-2}-\alpha^{2}\mathrm{e}^{C}(x+1)^{2\alpha-2}]\\
& -\frac{\alpha\,(x+1)^{\alpha-1}\,\mathrm{e}^{C}[-\log\,(1-\mathrm{e}^{C})]^{-(1/\sigma)-1}}{\sigma[1-\mathrm{e}^{C}]}-\frac{\alpha\,(x+1)^{\alpha-1}\mathrm{e}^{C}}{1-\mathrm{e}^{C}}\\
& +\frac{\alpha\,(\theta-1)(x+1)^{\alpha-1}[-\log\,\{1-\mathrm{e}^{C}\}]^{-(1/\sigma)-1}\mathrm{e}^{-[-\log\,(1-\mathrm{e}^{C})]^{-1/\sigma}}}{\sigma\,\mathrm{e}^{C}\,[1-\mathrm{e}^{C}][1-\mathrm{e}^{-\{-\log\,(1-\mathrm{e}^{C})\}^{-1/\sigma}}]}.
\end{aligned}
$$

Similarly, the critical points of the hrf of $X$ are obtained from the equation:

$$
\begin{aligned}
& \alpha^{-1}\mathrm{e}^{-C}(x+1)^{1-\alpha}[(\alpha-1)\,\alpha\,\mathrm{e}^{C}(x+1)^{\alpha-2}-\alpha^{2}\mathrm{e}^{C}(x+1)^{2\alpha-2}]\\
& +\frac{\alpha\,\mathrm{e}^{C}(x+1)^{\alpha-1}[-\log\,(1-\mathrm{e}^{C})]^{-\frac{1}{\sigma}-1}}{\sigma(1-\mathrm{e}^{C})}-\frac{\alpha\left(\frac{1}{\sigma}+1\right)\mathrm{e}^{C}(x+1)^{\alpha-1}}{[1-\mathrm{e}^{C}]\log\,(1-\mathrm{e}^{C})}\\
& -\frac{\alpha(\alpha-1)\,(x+1)^{\alpha-1}[-\log\,(1-\mathrm{e}^{C})]^{-(1/\sigma)-1}}{\sigma\,[1-\mathrm{e}^{C}][1-\mathrm{e}^{\{-\log\,(1-\mathrm{e}^{C})\}^{-1/\sigma}}]}+\frac{[1-\mathrm{e}^{(-\log\,\{1-\mathrm{e}^{C}\})^{-1/\sigma}}]^{\theta-1}}{\sigma\,\mathrm{e}^{-C}[1-\mathrm{e}^{C}][1-(1-\mathrm{e}^{(-\log\,\{1-\mathrm{e}^{C}\})^{-1/\sigma}})^{\theta}]}\\
& -\frac{\exp\,\{[-\log\,(1-\mathrm{e}^{C})]^{-1/\sigma}-C\}\alpha\,\theta(x+1)^{\alpha-1}[-\log\,(1-\mathrm{e}^{C})]^{-(1/\sigma)-1}\mathrm{e}^{[-\log\,(1-\mathrm{e}^{C})]^{-1/\sigma}}}{\sigma\,[1-\mathrm{e}^{C}][1-\mathrm{e}^{\{-\log\,(1-\mathrm{e}^{C})\}^{-1/\sigma}}]}.
\end{aligned}
$$

Some plots of the density of EGuNH for selected parameter values are presented in Figs 1–4 while plots of the hrf of EGuNH for random parameter values are presented (Figs 5–8). It is apparent that the density of EGuNH can be reversed-J, unimodal, and symmetrical. Similarly, EGuNH hazard rate shapes may tend to be increasing, decreasing, bathtub, or upside-down bathtub. The new model is much superior at fitting data sets in a variety of risk evaluation scenarios.

## 3.2 Central properties of EGuNH distribution

In this section, some useful expressions for the linear expansion, moments and incomplete moments of EGuNH distribution have been deduced using the Eq (16).

**Proposition 1**.

$$
f(x)=\sum_{m=0}^{\infty}\Pi_{m}\psi_{(x;\theta,\sigma,\alpha)}, \tag{26}
$$

where

$\psi_{(x;\theta,\sigma,\alpha)}=\alpha\,m(1+x)^{\alpha-1}\mathrm{e}^{1-(1+x)^{\alpha}}[1-\mathrm{e}^{1-(1+x)^{\alpha}}]^{m}$. Recalling the result defined in Eq (15) as

$$
F(x)=\sum_{m=1}^{\infty}\Pi_{m}(1-\mathrm{e}^{1-(1+x)^{\alpha}})^{m},
$$

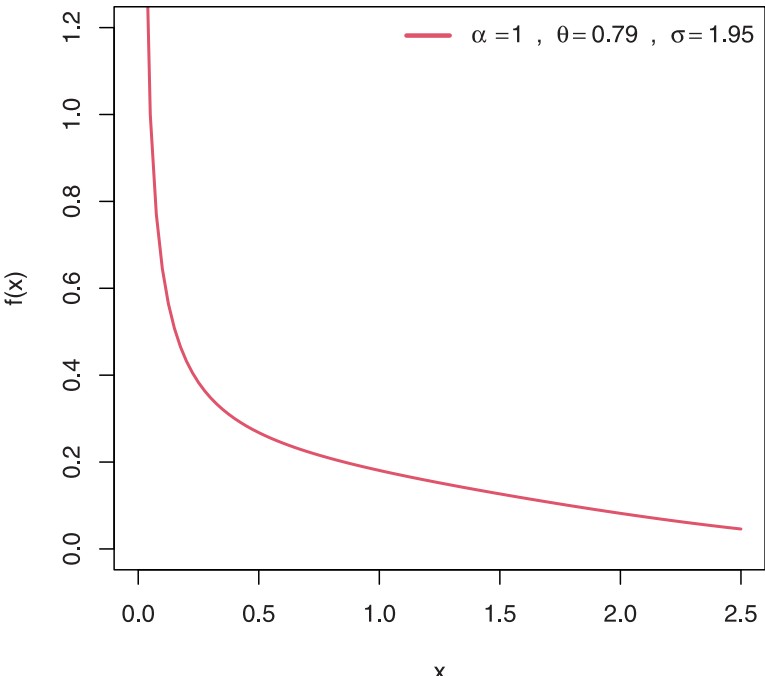

**Fig 1. Plots of EGuNH density for a variety of parameter combinations.**

A straightforward differentiation of the above result yields density by

$$f(x) = \sum_{m=1}^{\infty} \pi_m \psi_{(x;\theta,\sigma,\alpha)} \,, \tag{27}$$

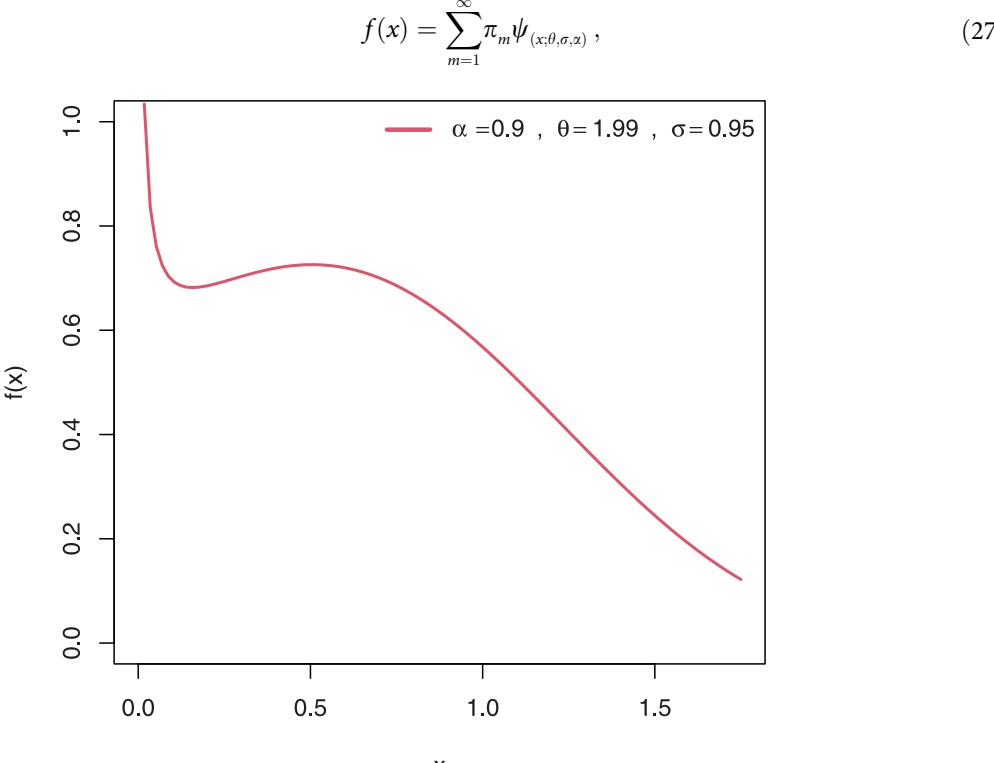

**Fig 2. Plots of EGuNH density for a variety of parameter combinations.**

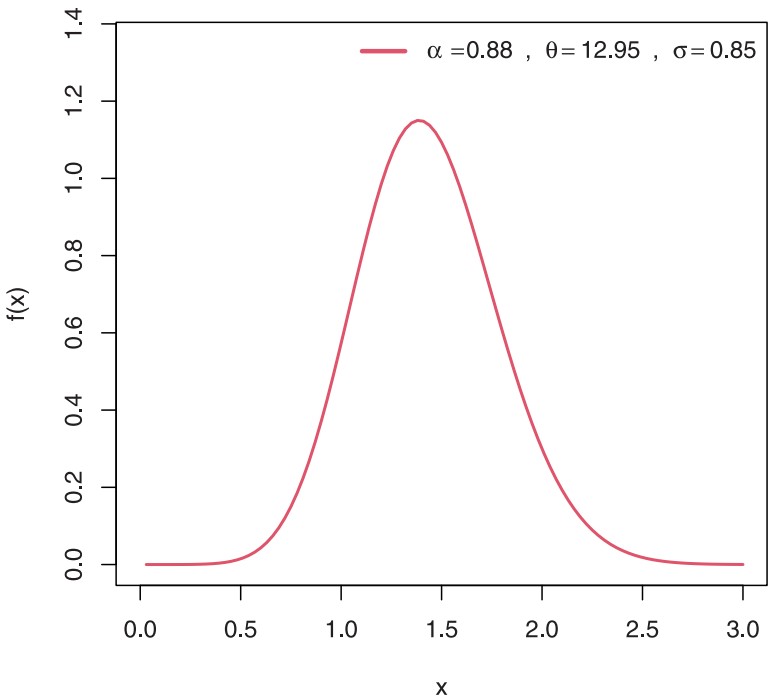

**Fig 3. Plots of EGuNH density for a variety of parameter combinations.**

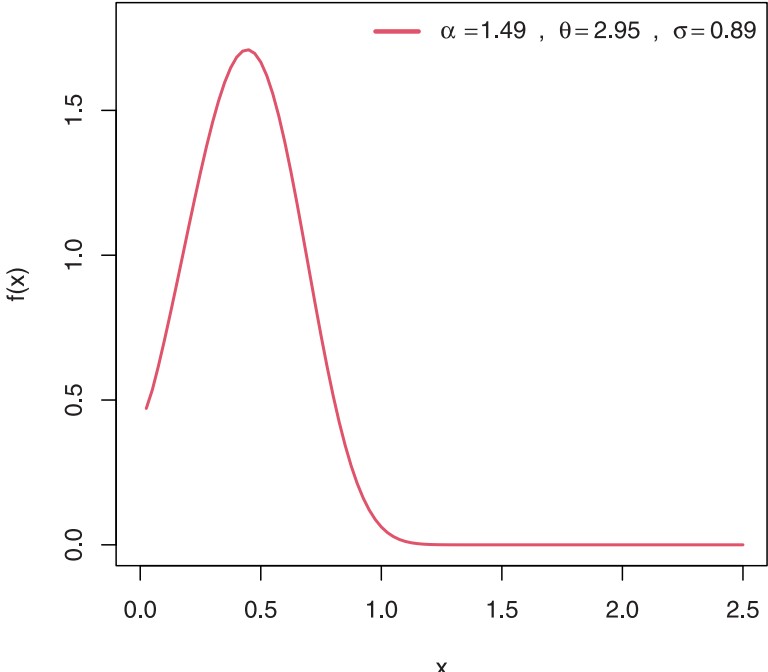

**Fig 4. Plots of EGuNH density for a variety of parameter combinations.**

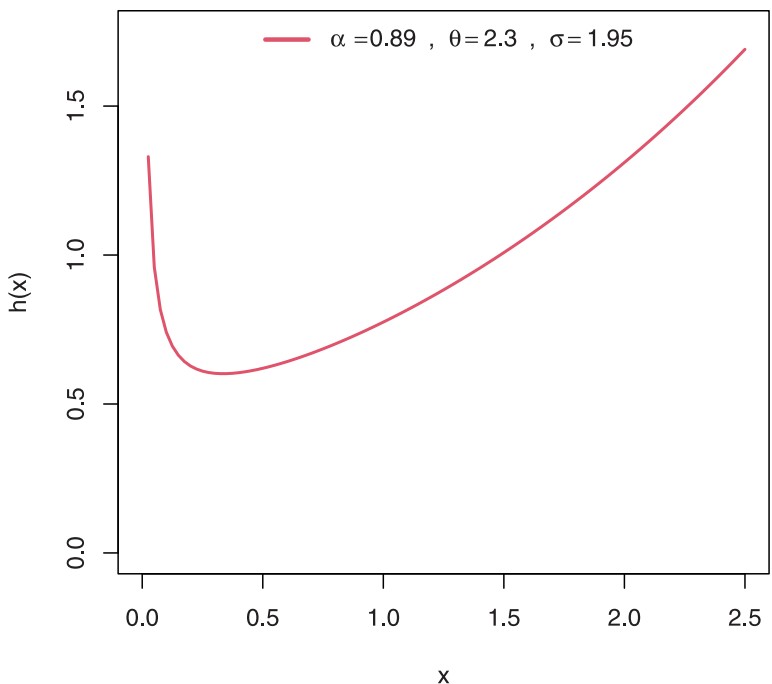

**Fig 5. Plots of EGuNH hazard rate for a variety of parameter combinations.**

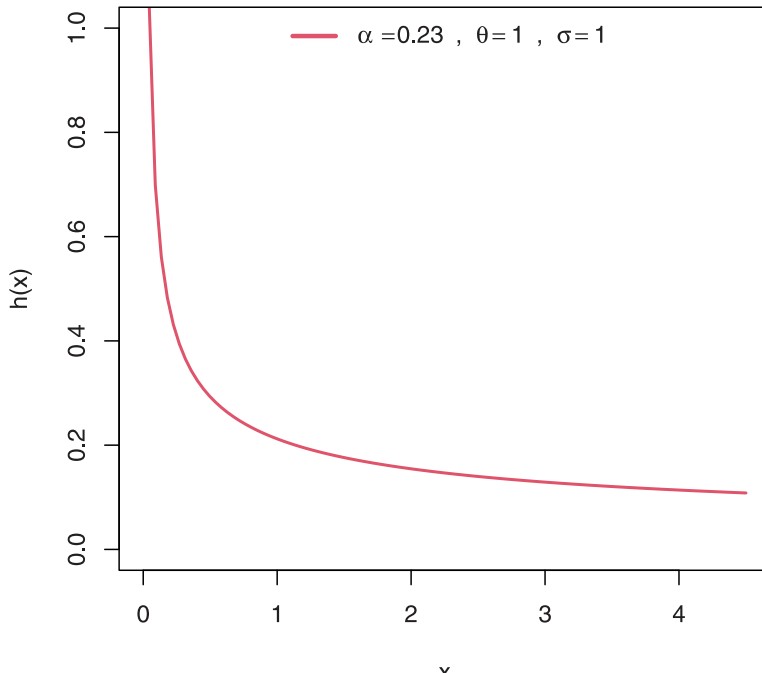

**Fig 6. Plots of EGuNH hazard rate for a variety of parameter combinations.**

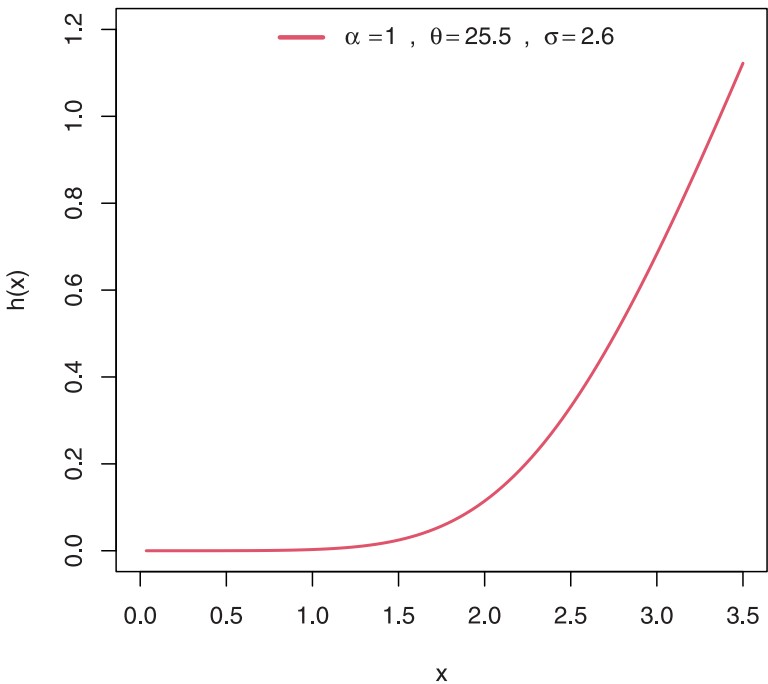

**Fig 7. Plots of EGuNH hazard rate for a variety of parameter combinations.**

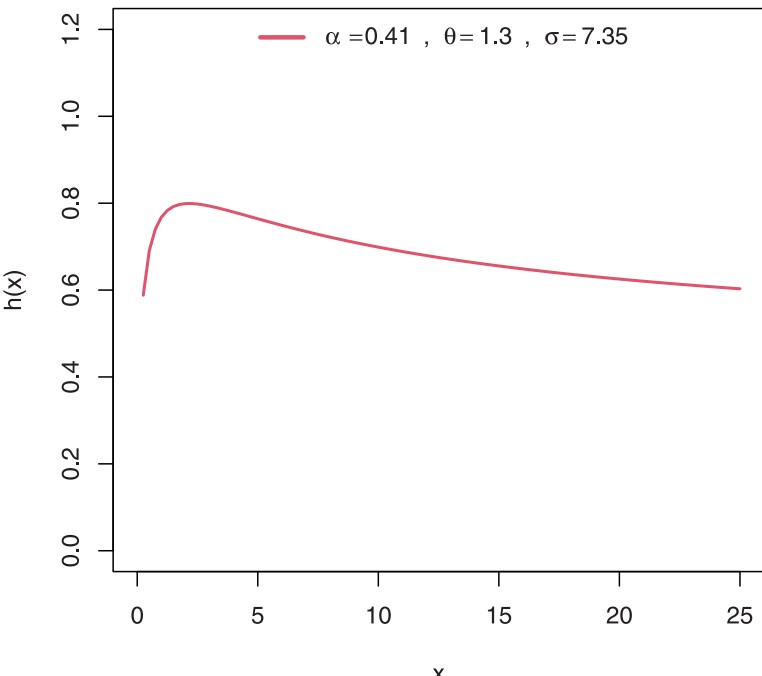

**Fig 8. Plots of EGuNH hazard rate for a variety of parameter combinations.**

The result in (27) is the linear expansion of NH densities. Hence, we shall derive several core properties of EGuNH using the major result of Eq 27.

**Proposition 2**.

Let $W$ be a av with density $\psi(x; m, \alpha)$. Then, several properties of $W$ can follow from those of $X$. The $s$th ordinary moment of $X$ can be written as

$$\mu'_s = \sum_{m=2}^{\infty} \sum_{p=0}^{m-1} m\, \Pi_m (-1)^{s+p+1} e^{p+1} \binom{m-1}{p} I(s, 0, p+1),$$
(28)

where $\Pi_m = (-1)^{m+2} \sum_{i,j=1}^{\infty} \sum_{k=0}^{\infty} \frac{(-1)^{i+j}\, j^i\, \Gamma[m-i/\sigma(k+1)]}{i!\, j!\, \Gamma[-i/\sigma(k+1)]} \binom{\theta}{\sigma} P_k(-i/\sigma)$ and

$$I(s, 0, p) = \sum_{l=0}^{s} (-1)^{s-l} \binom{s}{l} \gamma\left(\frac{l}{\alpha} + 1; p+1\right)$$

Utilizing the results derived in Eq (19), the $s$th moments are defined in (28).

**Proposition 3**.

The $s$th incomplete moment expression can be written as

$$m_s(W) = \sum_{m=2}^{\infty} \sum_{p=0}^{m-1} m\, \Pi_m (-1)^{s+p+1} e^{p+1} \binom{m-1}{p} Z(s, x),$$
(29)

where

$$Z(s, x) = \sum_{l=0}^{s} (-1)^{s-l} \binom{s}{l} \Gamma\left(\frac{l}{\alpha} + 1; p+1(1+x)^\alpha\right).$$

Following the results defined in Eq (21), the $s$th incomplete moments are defined in (29). The skewness $\gamma_1 = \kappa_3/\kappa_2^{3/2}$ and kurtosis $\gamma_2 = \kappa_4/\kappa_2^2$ of $X$ can be calculated from the third and fourth standardized cumulants. The classical skewness (Fig 9) and kurtosis plots (Fig 10) of the

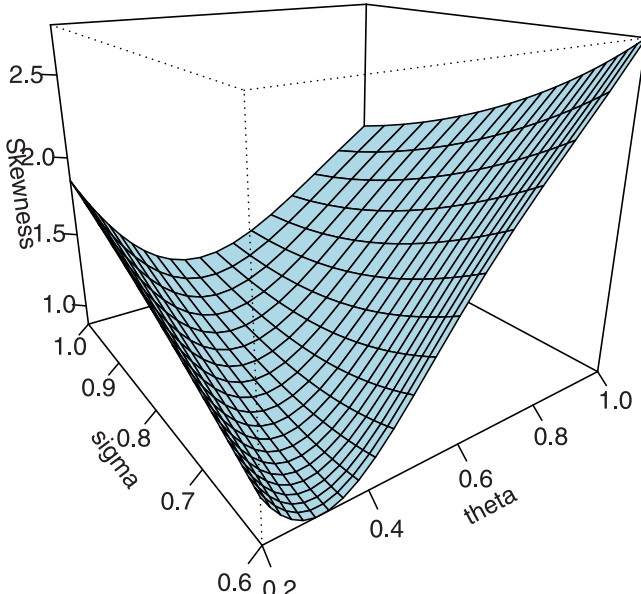

**Fig 9. Bowley skewness of EGuNH.**

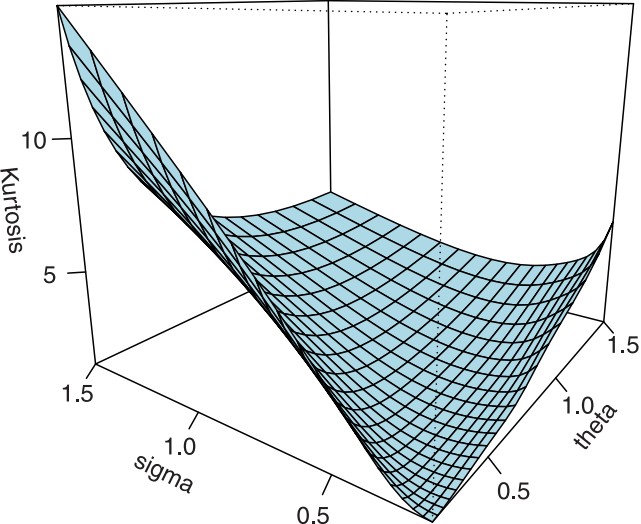

**Fig 10. Moors kurtosis of EGuNH.**

EGuNH distribution are displayed. Additionally, we provide the graphical illustration of MacGvillary skewness (MGs), which is based on quantile approach, in Figs 11 and 12. These plots reveal that the parameters $\theta$ and $\sigma$ play a decisive role in modeling the skewness and kurtosis behaviors of $X$.

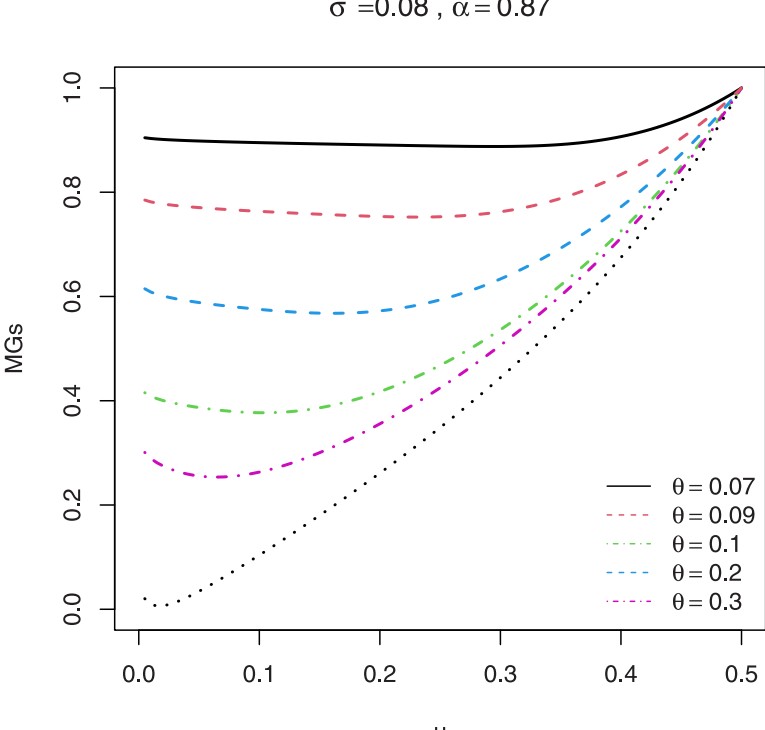

**Fig 11. MacGillivray skewness of EGuNH for a variety of parameter combinations.**

$$\theta = 0.1 \ , \ \alpha = 1.2$$

**Fig 12. MacGillivray skewness of EGuNH for a variety of parameter combinations.**

MacGillivray [35] proposed another method to evaluate the skewness measure based on the qf and is defined as

$$MGs = \frac{\rho_1(u; \theta, \sigma, \alpha)}{\rho_2(u; \theta, \sigma, \alpha)} = \frac{Q(1-u) + Q(u) - 2Q(1/2)}{Q(1-u) - Q(u)} ,$$

where $u \epsilon (0, 1)$, $Q(.)$ is the qf defined in (32).

The MG skewness plots are very sensitive for extremely small values of parameter $\theta$ and $\sigma$ which certainly signifies longer tails of EGuNH. Likewise, plots of the Lorenz (Fig 13) and Bonferroni (Fig 14) curves of EGuNH distribution for some random values are displayed. These plots reveal how the distribution parameters affect inequality measures which can be used to establish some orderings, an essential feature for applied statisticians. Some descriptive statistics related to EGuNH are presented in Tables 2 and 3, respectively.

## 3.3 Acturial measures EGuNH: Value at risk

The theory of finance is based upon risk evaluation. Investors are particularly interested to invest in entities in which there is minimum risk (specified with high degree of confidence) of losing money. In finance, value at risk (VaR) is the most extensively used metric for assessing liability. It is also known as quantile risk measure or quantile premium principle of the distribution of aggregate losses. It is characterised by a level of assurance "q" (usually at 95% and 99%). To a layman, VaR answers a simple question that "What is the worst case scenario that can happen in a particular investment?"

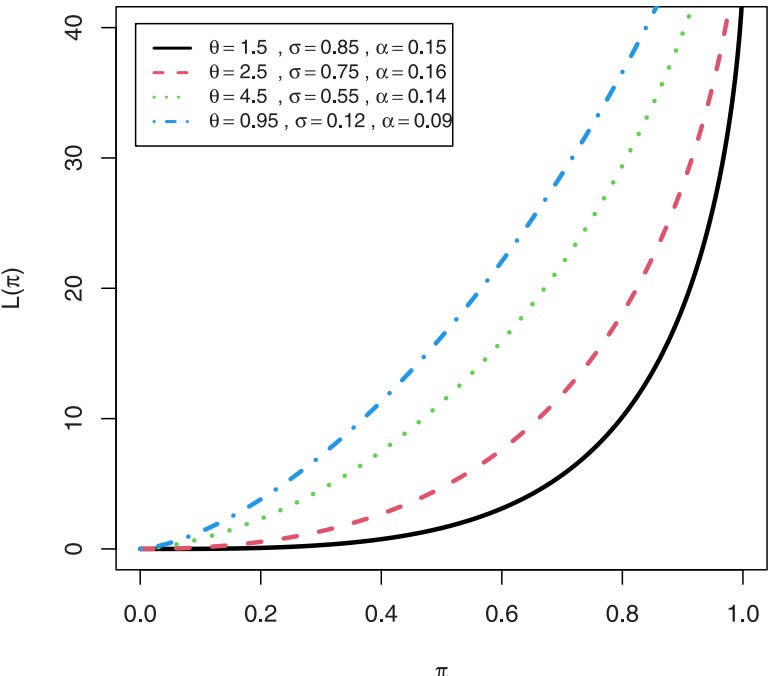

**Fig 13. Plots of the Lorenz curves of EGuNH distribution.**

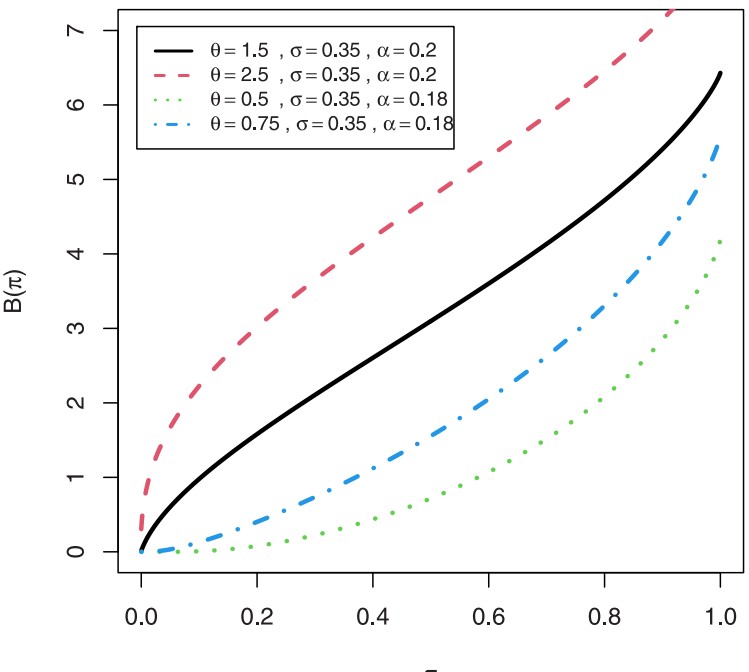

**Fig 14. Plots of the Bonferroni curves of EGuNH distribution.**

**Table 2. Descriptive measures of EGuNH for some parameter values.**

| Parameter values | Descriptives | | | | |
|---|---|---|---|---|---|
| $(\theta, \sigma, \alpha)$ | Q1 | Q2 | Q3 | B | M |
| (2.1, 0.1, 1.5) | 0.275 | 0.294 | 0.311 | −0.060 | 1.248 |
| (2.1, 0.29, 1.5) | 0.255 | 0.310 | 0.361 | −0.041 | 1.231 |
| (85.5, 0.979, 0.85) | 1.990 | 2.182 | 2.395 | 0.053 | 1.241 |
| (6.5, 0.33, 1.35) | 0.39 | 0.436 | 0.482 | −0.003 | 1.234 |
| (1.5, 1.9, 2.15) | 0.012 | 0.189 | 0.451 | 0.192 | 0.866 |
| (0.91, 2.1, 1.5) | 0 | 0.048 | 0.497 | 0.805 | 1.314 |
| (1.7, 9.1, 1.5) | 0 | 0.604 | 2.536 | 0.524 | 0.757 |
| (0.7, 5.1, 1.3) | 0 | 1.394 | 1.996 | 0.667 | 1.757 |
| (1.7, 0.1, 0.85) | 0 | 0.314 | 2.267 | 0.609 | 0.888 |

If $X$ has pdf (25), then VaR is the $q$th quantile of its cdf (24), defined as

$$VaR_q(x) = [1 - \log(1 - e^{-\{-\log(1-q^{1/\theta})\}^{-\sigma}})]^{1/\alpha} - 1 \tag{30}$$

## 3.4 Acturial measures EGuNH: Expected shortfall

Despite of the popularity of VaR measures, there are many shortcomings (see [36]). To counter inherent problems in VaR, Artzner et al. [37, 38] proposed the use of expected shortfall (ES). Expected shortfall quantifies the average loss in states beyond the VaR level. ES has a number of aliases such as "conditional VaR", "mean excess loss" or "tail VaR". We define the ES as follows

$$
\begin{aligned}
ES_q(x) &= E[X|X \geq VaR_q(x)] \\
ES_q(x) &= \frac{1}{q}\int_0^q \left([1 - \log(1 - e^{-\{-\log(1-q^{1/\theta})\}^{-\sigma}})]^{1/\alpha} - 1\right)dx.
\end{aligned}
\tag{31}
$$

For a combination of various parameter values, plots of VaRs (Fig 15) and ESs (Fig 16) are displayed respectively.

**Table 3. Moments and moment ratios of EGuNH for some parameter combinations.**

| Parameter values | Moments and moments ratio | | | | | | | | |
|---|---|---|---|---|---|---|---|---|---|
| $(\theta, \sigma, \alpha)$ | $E(x)$ | $E(x^2)$ | $E(x^3)$ | $E(x^4)$ | $V(x)$ | $\sigma(x)$ | CV | CS | CK |
| (0.83, 2.1, 1.5) | 0.225 | 0.233 | 0.241 | 0.299 | 0.141 | 0.401 | 1.322 | 1.589 | 4.253 |
| (0.91, 2.1, 1.5) | 0.295 | 0.263 | 0.291 | 0.365 | 0.176 | 0.420 | 1.422 | 1.680 | 4.944 |
| (1.7, 9.1, 1.5) | 1.343 | 4.258 | 15.684 | 63.478 | 2.456 | 1.567 | 1.167 | 1.680 | 4.944 |
| (2.1, 0.1, 1.5) | 0.292 | 0.086 | 0.026 | 0.008 | 0.001 | 0.027 | 0.094 | 2.505 | 6.661 |
| (2.1, 0.29, 1.5) | 0.306 | 0.100 | 0.034 | 0.012 | 0.006 | 0.078 | 0.254 | -0.285 | 2.957 |
| (1.5, 1.9, 2.15) | 0.597 | 0.767 | 1.219 | 2.205 | 0.410 | 0.640 | 1.072 | 88.455 | 47.815 |
| (6.5, 0.33, 1.35) | 0.436 | 0.194 | 0.089 | 0.041 | 0.005 | 0.068 | 0.157 | -0.044 | 3.206 |
| (85.5, 0.979, 0.85) | 2.163 | 4.764 | 10.965 | 24.462 | 0.088 | 0.297 | 0.137 | 0.437 | 3.289 |

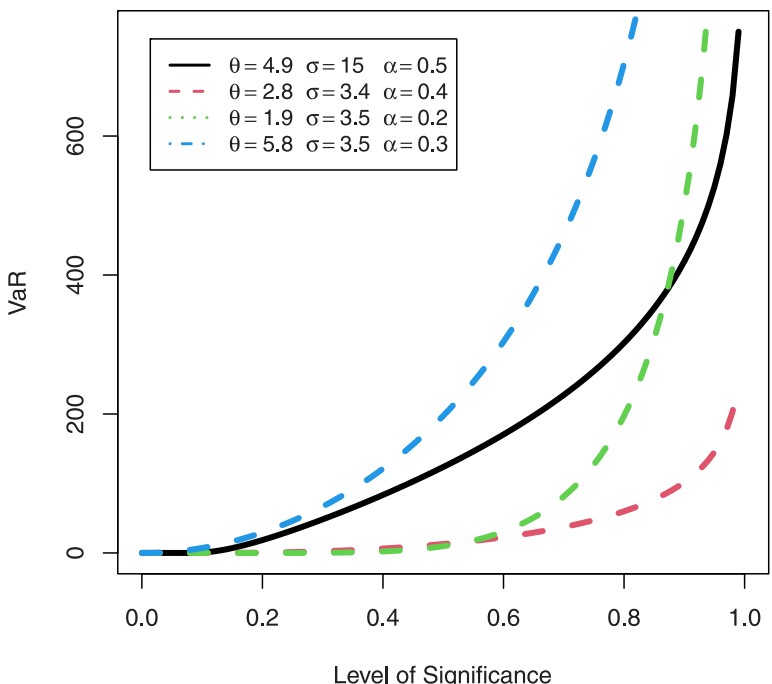

**Fig 15. Plots of the VaR of EGuNH distribution for some random parameter values.**

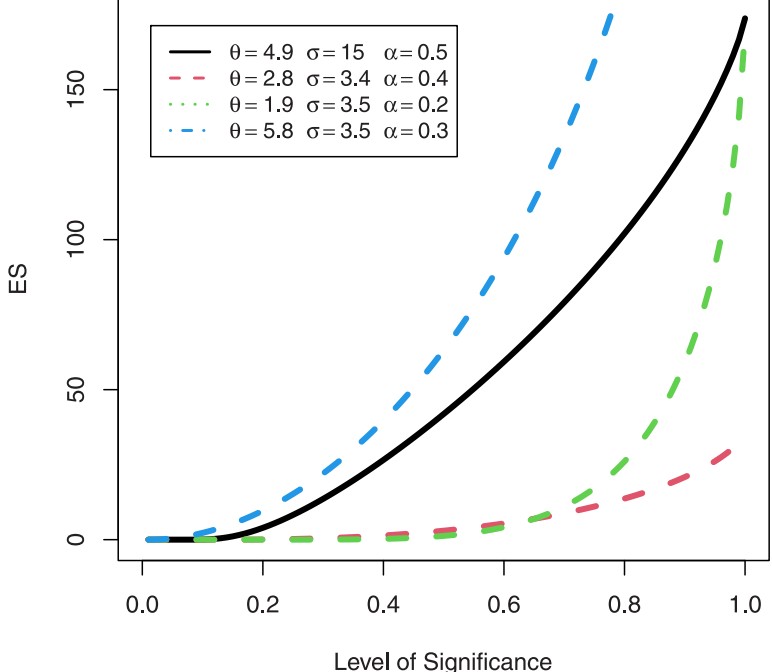

**Fig 16. Plots of the ES of EGuNH distribution for some random parameter values.**

### 3.5 Parameter estimation of EGuNH

The log-likelihood function $\Im$ for the vector of parameters $\Theta = (\theta, \sigma, \alpha)^{\top}$ for the model defined in (25) is given by

$$n \log(\alpha) + n \log(\theta) - n \log(\sigma) + (\theta - 1) \sum_{i=1}^{n} \log \left[ 1 - e^{\{-\log(1-e^{1-(x_i+1)^{\alpha}})\}^{-1/\sigma}} \right]$$

$$-\sum_{i=1}^{n} [-\log\{1 - e^{1-(x_i+1)^{\alpha}}\}]^{-1/\sigma} - \frac{(\sigma+1)}{\sigma} \sum_{i=1}^{n} \log\left[-\log\{1 - e^{1-(x_i+1)^{\alpha}}\}\right]$$

$$+\sum_{i=1}^{n}[1 - (x_i + 1)^{\alpha}] + (\alpha - 1)\sum_{i=1}^{n} \log(x_i + 1),$$

The components of the score vector $U(\Theta)$ are

$$U_{\theta} = \frac{n}{\theta} + \sum_{i=1}^{n} \log\left[1 - e^{\{-\log\{1-e^{1-(x_i+1)^{\alpha}}\}\}^{-1/\sigma}}\right],$$

$$U_{\sigma} = \left(\frac{\sigma+1}{\sigma}\right) \sum_{i=1}^{n} \log\left[-\log\{1 - e^{1-(x_i+1)^{\alpha}}\}\right] - \sigma^{-1} \sum_{i=1}^{n} \log\left[-\log\left(1 - e^{1-(x_i+1)^{\alpha}}\right)\right] - \frac{n}{\sigma}$$

$$+ \sigma^{-2} \sum_{i=1}^{n} \log\left[-\log\{1 - e^{1-(x_i+1)^{\alpha}}\}\right][-\log\{1 - e^{1-(x_i+1)^{\alpha}}\}]^{-1/\sigma}$$

$$+ (\theta - 1)\sum_{i=1}^{n} -\log\left[-\log\{1 - e^{1-(x_i+1)^{\alpha}}\}\right]e^{\{-\log(1-e^{1-(x_i+1)^{\alpha}})\}^{-1/\sigma}}$$

$$\times [-\log\{1 - e^{1-(x_i+1)^{\alpha}}\}]^{-1/\sigma} \sigma^2 [1 - e^{\{-\log(1-e^{1-(x_i+1)^{\alpha}})\}^{-1/\sigma}}],$$

$$U_{\alpha} = \frac{n}{\alpha} + \sum_{i=1}^{n} (x_i + 1)^{\alpha} [-\log(x_i + 1)] - \left(\frac{\sigma+1}{\sigma}\right)\sum_{i=1}^{n} \frac{e^{1-(x_i+1)^{\alpha}}(x_i+1)^{\alpha}\log(x_i+1)}{[1 - e^{1-(x_i+1)^{\alpha}}]\log[1 - e^{1-(x_i+1)^{\alpha}}]}$$

$$+\sum_{i=1}^{n} \log(x_i + 1) - \sum_{i=1}^{n} \frac{e^{1-(x_i+1)^{\alpha}}(x_i+1)^{\alpha}\log(x_i+1)[-\log\{1 - e^{1-(x_i+1)^{\alpha}}\}]^{-\frac{1}{\sigma}-1}}{\sigma[1 - e^{1-(x_i+1)^{\alpha}}]}$$

$$+(\theta - 1)\sum_{i=1}^{n} (x_i + 1)^{\alpha}[-\log(x_i + 1)][-\log\{1 - e^{1-(x_i+1)^{\alpha}}\}]^{-\frac{1}{\sigma}-1}$$

$$\times \exp\left[\{-\log(1 - e^{1-(x_i+1)^{\alpha}})\}^{-1/\sigma} - (x_i + 1)^{\alpha} + 1\right]$$

$$\times \{\sigma[1 - e^{1-(x_i+1)^{\alpha}}][1 - e^{(-\log\{1-e^{1-(x_i+1)^{\alpha}}\})^{-1/\sigma}}]\}.$$

The MLE $\hat{\Theta}$ of $\Theta$ can also be obtained by solving the nonlinear equations $U_{\theta} = 0$, $U_{\sigma} = 0$ and $U_{\alpha} = 0$. Because these equations cannot be solved analytically, the estimates can be calculated numerically using statistical software.

### 3.6 Simulation study of EGuNH distribution

The qf of the EGuNH distribution has an explicit form as follows

$$Q(u) = [1 - \log(1 - e^{-\{-\log(1-u^{1/\theta})\}^{-\sigma}})]^{1/\alpha} - 1. \tag{32}$$

Here, we use Monte Carlo simulations to demonstrate the performance and correctness of maximum likelihood estimations of the EGuNH parameters by inverting Eq (24) to generate a sample data from the model. The simulation study is perform for sample sizes $n = 50, 100, 200, 500$, and parameter combinations: I: $\theta = 0.2$, $\sigma = 0.75$ and $\alpha = 0.5$, II: $\theta = 2.2$, $\sigma = 0.45$ and $\alpha =$

**Table 4. AEs, Biases, MSEs and CPs for combination-I.**

| | | $n = 50$ | | | | $n = 100$ | | | |
|---|---|---|---|---|---|---|---|---|---|
| | AEs | Bias | MSEs | CPs | AEs | Bias | MSEs | CPs |
| $\theta$ | 0.279 | 0.081 | 0.447 | 0.99 | 0.274 | 0.074 | 0.436 | 0.98 |
| $\sigma$ | 0.927 | -1.661 | 0.691 | 1.00 | 0.895 | -1.651 | 0.683 | 0.97 |
| $\alpha$ | 0.925 | -0.241 | 0.494 | 0.98 | 0.817 | -0.203 | 0.441 | 0.92 |
| | | $n = 200$ | | | | $n = 500$ | | | |
| | AEs | Bias | MSEs | CPs | AEs | Bias | MSEs | CPs |
| $\theta$ | 0.272 | 0.072 | 0.429 | 0.96 | 0.272 | 0.051 | 0.297 | 0.94 |
| $\sigma$ | 0.784 | -1.350 | 0.669 | 0.95 | 0.757 | -1.115 | 0.662 | 0.95 |
| $\alpha$ | 0.629 | -0.220 | 0.359 | 0.95 | 0.548 | -0.204 | 0.320 | 0.96 |

0.5, III: $\theta = 3.4$, $\sigma = 0.75$ and $\alpha = 1.5$ and IV: $\theta = 3.4$, $\sigma = 1.35$ and $\alpha = 1.5$. This study is carried out for $N = 2000$ times, each with given $n$ and computed the average estimates (AEs) as well as their average biases (Bias), mean squared errors (MSEs) and coverage probabilities (CPs) of the MLEs.

$$Bias(\hat{\theta}) = \sum_{i=1}^{N} \frac{\hat{\theta}_i}{N} - \theta,$$

$$MSE(\hat{\theta}) = \sum_{i=1}^{N} \frac{(\hat{\theta}_i - \theta)^2}{N}$$

$$CPs(\hat{\theta}) = \sum_{i=1}^{N} \frac{[\{\hat{\theta}_i - (1.95996 \times SE_{\hat{\theta}_i})\}, \{\hat{\theta}_i + (1.95996 \times SE_{\hat{\theta}_i})\}]}{N}$$

The AEs, Bias, MSEs and CPs for the parameters $\theta$, $\sigma$ and $\alpha$ are given in Tables (4)–(7). The empirical findings suggests that the bias and MSEs decreases as sample size increases. Further, the empirical CPs are quite close to the nominal level of 95%. As a result, MLEs and their approximate findings can be used to evaluate and build approximated confidence intervals of the EGuNH distribution parameters $\theta$, $\sigma$ and $\alpha$.

**Table 5. AEs, Biases, MSEs and CPs for combination-II.**

| | | $n = 50$ | | | | $n = 100$ | | | |
|---|---|---|---|---|---|---|---|---|---|
| | AEs | Bias | MSEs | CPs | AEs | Bias | MSEs | CPs |
| $\theta$ | 2.307 | -0.031 | 0.027 | 1.00 | 2.197 | -0.023 | 0.012 | 0.92 |
| $\sigma$ | 0.499 | -0.021 | 0.011 | 0.91 | 0.495 | -0.005 | 0.002 | 0.98 |
| $\alpha$ | 0.511 | 0.018 | 0.018 | 0.99 | 0.510 | 0.010 | 0.005 | 0.93 |
| | | $n = 200$ | | | | $n = 500$ | | | |
| | AEs | Bias | MSEs | CPs | AEs | Bias | MSEs | CPs |
| | AEs | Bias | MSEs | CPs | AEs | Bias | MSEs | CPs |
| $\theta$ | 2.196 | -0.014 | 0.002 | 0.97 | 2.198 | -0.011 | 0.001 | 0.95 |
| $\sigma$ | 0.501 | 0.001 | 0.001 | 0.95 | 0.498 | 0.001 | 0.001 | 0.94 |
| $\alpha$ | 0.508 | 0.008 | 0.003 | 0.94 | 0.501 | 0.000 | 0.002 | 0.95 |

**Table 6. AEs, Biases, MSEs and CPs for combination-IV.**

|  |  | *n* = 50 |  |  |  | *n* = 100 |  |  |
|---|---|---|---|---|---|---|---|---|
|  | AEs | Bias | MSEs | CPs | AEs | Bias | MSEs | CPs |
| $\theta$ | 3.507 | -0.201 | 0.027 | 0.98 | 3.405 | -0.123 | 0.019 | 0.93 |
| $\sigma$ | 0.769 | 0.021 | 0.015 | 0.97 | 0.761 | 0.010 | 0.013 | 0.99 |
| $\alpha$ | 2.541 | 0.058 | 0.023 | 0.90 | 2.130 | 0.030 | 0.018 | 0.97 |
|  |  | *n* = 200 |  |  |  | *n* = 500 |  |  |
|  | AEs | Bias | MSEs | CPs | AEs | Bias | MSEs | CPs |
|  | AEs | Bias | MSEs | CPs | AEs | Bias | MSEs | CPs |
| $\theta$ | 3.405 | -0.014 | 0.010 | 0.96 | 3.401 | -0.011 | 0.007 | 0.96 |
| $\sigma$ | 0.755 | 0.005 | 0.006 | 0.94 | 0.753 | 0.003 | 0.004 | 0.95 |
| $\alpha$ | 1.615 | 0.015 | 0.004 | 0.96 | 1.511 | 0.011 | 0.002 | 0.94 |

**Table 7. AEs, Biases, MSEs and CPs for combination-IV.**

|  |  | *n* = 50 |  |  |  | *n* = 100 |  |  |
|---|---|---|---|---|---|---|---|---|
|  | AEs | Bias | MSEs | CPs | AEs | Bias | MSEs | CPs |
| $\theta$ | 3.877 | -0.281 | 0.097 | 0.97 | 3.595 | -0.223 | 0.019 | 0.92 |
| $\sigma$ | 0.889 | 0.021 | 0.025 | 1.00 | 0.827 | 0.010 | 0.020 | 0.94 |
| $\alpha$ | 2.541 | 0.058 | 0.033 | 0.97 | 2.230 | 0.030 | 0.028 | 0.99 |
|  |  | *n* = 200 |  |  |  | *n* = 500 |  |  |
|  | AEs | Bias | MSEs | CPs | AEs | Bias | MSEs | CPs |
| $\theta$ | 3.495 | -0.019 | 0.010 | 0.96 | 3.410 | -0.013 | 0.009 | 0.95 |
| $\sigma$ | 0.765 | 0.005 | 0.016 | 0.97 | 0.751 | 0.003 | 0.008 | 0.96 |
| $\alpha$ | 1.915 | 0.015 | 0.014 | 0.95 | 1.507 | 0.011 | 0.010 | 0.95 |

## 4 Applications of the EGuNH distribution

Statistical methods that fail to account for all of the uncertainties in the model are prone to produce an overly optimistic assessment of future extremes, are frequently contradicted by observed extreme events in a variety of scientific fields. The current literature regarding extreme value theory is full of such models in which data sets are meteorology data such as earthquakes, floods, rains, droughts, hurricanes etc. (see [15–32]). On the contrary, health hazards is an area of extreme value theory which should be explored. Death, damage, or disease; exacerbation of underlying medical disorders; and negative effects on mental health are some of the health hazards of climate-related increases in exposure to extreme occurrences.

In this section, we provide some applications of the EGuNH model on three real life phenomenons, two of which related to health hazards in extreme value theory. We estimate the unknown parameters of the distributions by the principal of maximum likelihood. We compute the log-likelihood function evaluated at the MLEs ($-\hat{\ell}$) using the method of a limited-memory quasi-Newton code for bound-constrained maximization (L-BFGS-B). In order to select the best probability model, a variety of criteria for evaluating information (ICs) can be considered. We considered the following well-known ICs: the maximized log-likelihood ($-\hat{\ell}$), Akaike Information criterion (AIC), Anderson-Darling ($A^\star$), Cramér-von Mises ($W^\star$) and Kolmogorov-Smirnov measures ($D^\star$; P-value ($p^\star$)), where lower values of all these statistics except higher $p^\star$ values of K-S, indicate good fits. The required computations are carried out using the R script `AdequacyModel` which is freely accessible from http://cran.r-project.org/web/packages/AdequacyModel/AdequacyModel.pdf.

**Table 8. The comparative fitted models.**

| Distribution | Author(s) |
|---|---|
| GaNH | Cordiero et al., (2015) [31] |
| LxNH | Ramirez et al., (2020) [39] |
| TLNH | Yuwadee Sangsanit and Winai Bodhisuwan, (2016) [40] |
| ENH | Lemonte et al., (2013) [41] |
| MONH | Lemonte et al., (2016) [42] |
| NH | Nadarajah and Haghighi, (2011) [43] |

The fits of the EGuNH distribution is compared with other competitive models which are given in Table 8. The parameters are all positive real numbers of these densities.

## 4.1 Meteorology data

Meteorological phenomena are weather events that most individuals are affected by, due to changes in extreme weather and climatic events, such as earthquakes, heat waves, floods, hurricanes, droughts etc. The present data is taken from [44], denoted by D1, gives the time in days between successive serious earthquakes world-wide. An earthquake is included if its magnitude was at least 7.5 on the Richter scale, or if over 1000 people were killed. There were 63 earthquakes recorded altogether, and so 62 recorded waiting times. The data are: 840, 157, 145, 44, 33, 121, 150, 280, 434, 736, 584, 887, 263, 1901, 695, 294, 562, 721,40, 1336, 335, 1354, 454,139, 780, 203, 436, 30, 246, 1617, 638, 937, 735,76, 710, 36, 667,384, 129, 46, 402, 194, 40, 556, 99, 9, 209, 599, 38, 365, 92, 82, 220, 759, 304, 83, 319, 375, 832, 460, 567, 328.

## 4.2 Cancer data

According to [45], extreme events have the potential to disrupt the delivery of cancer care. For example, some deadly carcinogens may be released into communities as a result of hurricanes and wild fires; industry shutdowns may result in a shortage of life-saving medical equipment in hospitals, causing shortages in cancer facilities across the country; and infrastructure collapse may limit access to patients undergoing cancer therapies. The following two data, denoted by cancer 1 (D2) and cancer 2 (D3) are related to cancer patients.

**Cancer data 1**. The survival times, in weeks, of 33 patients who succumbed to Acute Myelogenous Leukemia are the subject of D2. This data was recently studied by the authors in [46] The data are: 65, 156, 100, 134, 16, 108, 121, 4,39, 143, 56, 26, 22, 1, 1, 5, 65, 56, 65, 17, 7, 16, 22, 3, 4, 2, 8, 4, 3, 30, 4,43.

**Cancer data 2**. D3 signifies the number of patients suffering from blood cancer. The Saudi Cancer Registry (SCR) provides such information, covering the time period from 1994 to the present day. The data is extracted from a report [47] which concerns an overview of cancer incidence statistics for Saudi Arabia in 2012. The data are: 1277, 1290, 1357, 1369, 1408, 1455, 1478, 1549, 115, 181, 255, 418, 441, 461, 516, 739, 743, 789, 807, 865, 924, 983, 1024, 1062, 1063, 1165, 1191, 1222, 1222, 1251, 1578, 1578, 1599, 1603, 1605, 1696, 1735, 1799, 1815, 1852. The descriptive statistics for each of the three data sets are given in Table 9.

**Table 9. The descriptive statistics related to D1, D2 and D3.**

| Data | Sample Size | Arithmetic Mean | Standard Deviation | Lowest | Highest | Skewness | Kurtosis |
|---|---|---|---|---|---|---|---|
| 1 | 62 | 437.21 | 399.93 | 9 | 1901 | 1.50 | 2.52 |
| 2 | 32 | 42.07 | 46.95 | 1 | 156 | 1.12 | 0.03 |
| 3 | 40 | 1137 | 481.60 | 115 | 1852 | -0.49 | -0.73 |

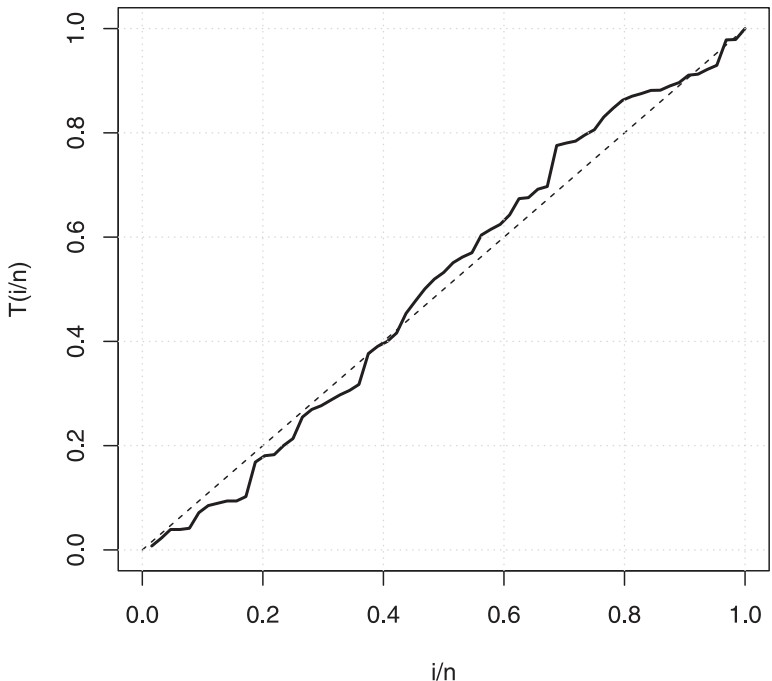

**Fig 17. TTT plot for D1.**

The empirical findings of all the three data are suggestive of the heavy tailed data. The TTT plots Figs (17)–(19) for the data sets are given. In particular, the TTT plots show bathtub, increasing and decreasing hrf, allowing us to fit EGuNH model on these data sets. The approximated hrf Figs (20)–(22) for each data point correlates to the TTT graphs. Table 10 summarizes

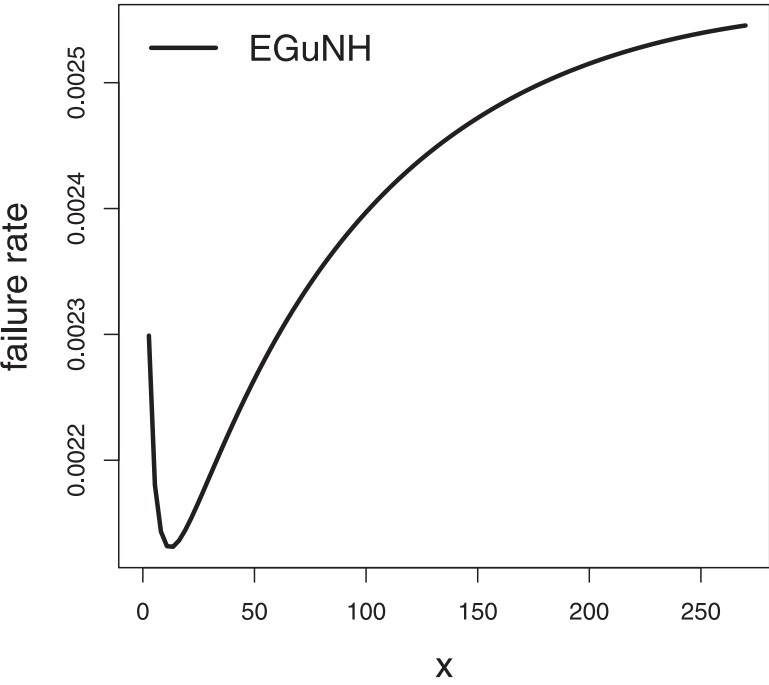

**Fig 18. TTT plot for D2.**

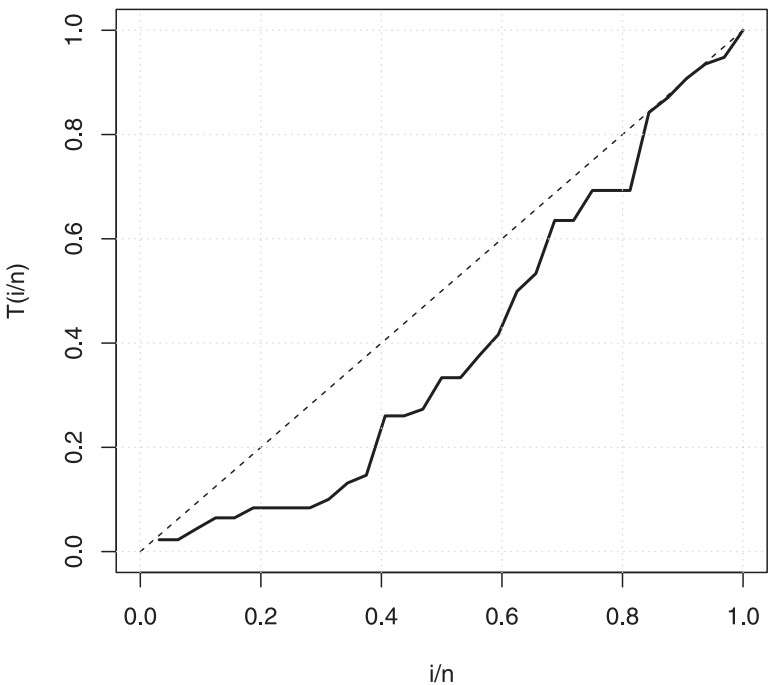

**Fig 19. TTT plot for D3.**

the results of the MLEs and their related standard errors (in parentheses) of the model parameters for the proposed model while the ICs are listed in Table 11 for the D1, D2 and D3, respectively. It is customary to supplement the analytical result defined in Tables 9 and 10, by displaying it graphically. Hence, the estimated pdfs Figs (23)–(25), PP–plots Figs (26)–(28),

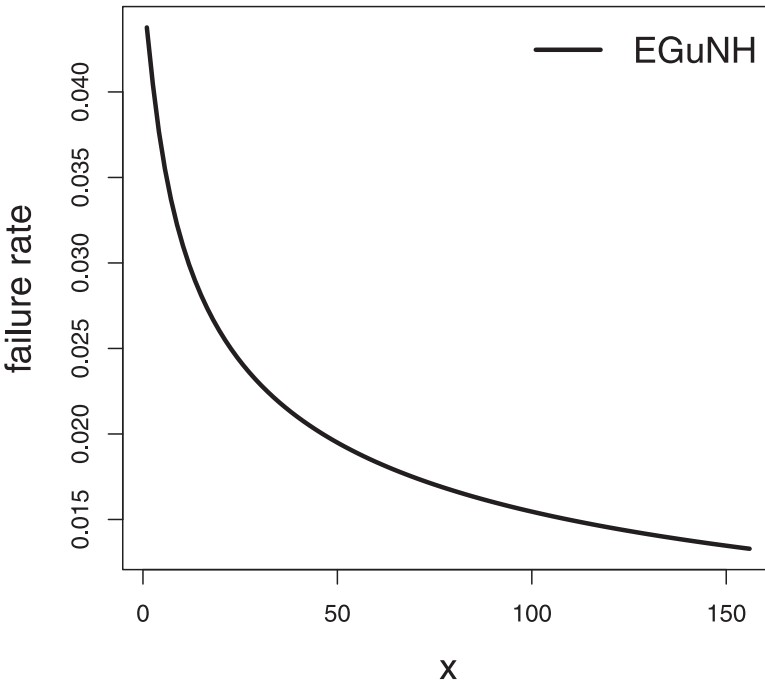

**Fig 20. Estimated hrf for D1.**

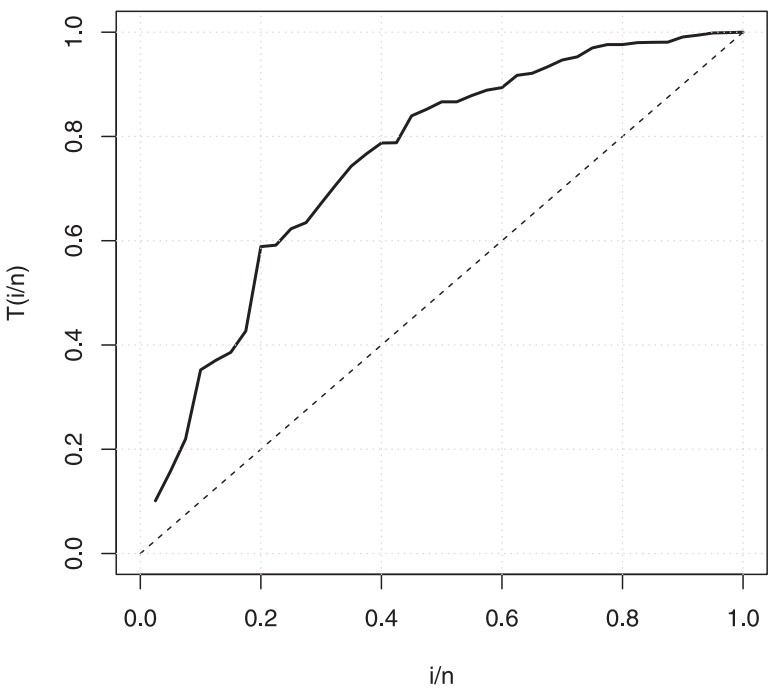

**Fig 21. TTT plot for D2.**

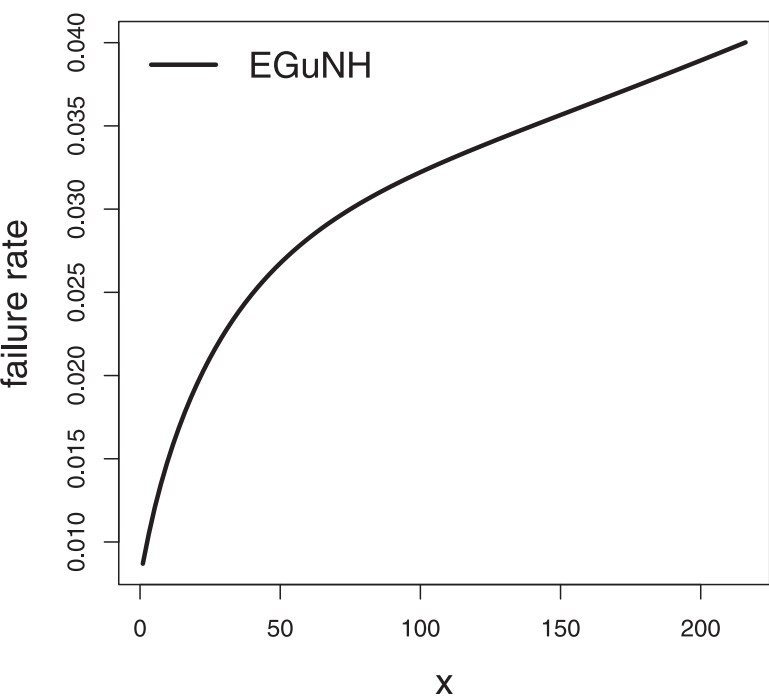

**Fig 22. Estimated hrf plots for D3.**

**Table 10. MLEs with their respective SEs (in parenthesis) for D1, D2 and D3.**

|  | Data 1 | Data 2 | Data 3 |
|---|---|---|---|
| Distribution | MLEs | MLEs | MLEs |
| EGuNH | 3.655,11.787,0.275 | 0.071,0.031,0.076 | 22.753,20.069,0.469 |
| $(\theta,\sigma,\alpha)$ | (2.217),(2.371),(0.073) | (0.015),(0.003),(0.001) | (5.093),(2.193),(0.011) |
| ENH | 1.285,0.764,0.004 | 1.256,1.716,0.011 | 4.515,0.971,0.002 |
| $(\theta,\alpha,\lambda)$ | (0.229),(0.117),(0.001) | (0.178),(0.556),(0.005) | (0.987),(0.066),(0.002) |
| GaNH | 0.295,7.164,3.874 | 1.6,1.256,0.013 | 1.146,3.802,0.002 |
| $(\theta,a\alpha,\lambda)$ | (1.018),(6.192),(2.987) | (0.542),(0.174),(0.007) | (0.094),(0.615),(0.518) |
| LxNH | 8.918,0.080,18.577 | 1.470,3.660,0.006 | 13.499,0.111,0.456 |
| $(\theta,\alpha,\lambda)$ | (3.211),(0.027),(58.410) | (0.126),(0.930),(0.002) | (3.672),(0.025),(0.643) |
| TLNH | 1.499,0.600,0.003 | 1.374,1.5490.007 | 11.215,0.371,0.009 |
| $(\theta,\alpha,\lambda)$ | (0.262),(0.084),(0.987) | (0.173),(0.405),(0.003) | (6.699),(0.072),(0.017) |
| MONH | 5.162,0.886,0.003 | 3.067,0.962,0.038 | 42.021,0.827,0.005 |
| $(\theta,\alpha,\lambda)$ | (0.559),(0.131),(0.001) | (0.159),(0.560),(0.055) | (2.127),(0.081),(0.311) |
| NH | 1.054,0.212 | 2.616,0.006 | 0.067,158.079 |
| $(\alpha,\lambda)$ | (0.104),(0.298) | (0.728),(0.012) | (0.095),(89.711) |

**Table 11. The statistics $-\hat{\ell}$, AIC, BIC, $A^\star$, $W^\star$, $D^\star$ and $p^\star$ for D1, D2 and D3.**

| Distribution | $-\hat{\ell}$ | AIC | BIC | $A^\star$ | $W^\star$ | $D^\star$ | $p^\star$ |
|---|---|---|---|---|---|---|---|
| Data set 1 |  |  |  |  |  |  |  |
| EGuNH | 441.19 | 889.90 | 896.33 | 0.41 | 0.05 | 0.08 | 0.79 |
| GaNH | 444.78 | 895.56 | 901.94 | 1.05 | 0.17 | 0.11 | 0.54 |
| LxNH | 448.86 | 903.71 | 910.14 | 1.58 | 0.25 | 0.18 | 0.45 |
| TLNH | 442.73 | 891.24 | 897.67 | 0.61 | 0.09 | 0.10 | 0.68 |
| ENH | 442.67 | 890.98 | 897.74 | 0.46 | 0.07 | 0.09 | 0.77 |
| MONH | 441.90 | 890.14 | 897.01 | 0.34 | 0.05 | 0.09 | 0.78 |
| NH | 442.27 | 891.75 | 897.15 | 0.31 | 0.06 | 0.08 | 0.78 |
| Data set 2 |  |  |  |  |  |  |  |
| EGuNH | 149.027 | 304.054 | 308.451 | 0.567 | 0.08 | 0.111 | 0.817 |
| GaNH | 150.203 | 306.406 | 310.803 | 0.588 | 0.09 | 0.122 | 0.729 |
| LxNH | 152.672 | 311.345 | 315.741 | 0.860 | 0.14 | 0.126 | 0.687 |
| TLNH | 150.104 | 306.205 | 310.602 | 0.565 | 0.08 | 0.127 | 0.681 |
| ENH | 150.101 | 306.202 | 310.599 | 0.581 | 0.08 | 0.126 | 0.692 |
| MONH | 150.036 | 306.071 | 310.468 | 0.593 | 0.09 | 0.143 | 0.534 |
| NH | 152.875 | 310.751 | 312.005 | 0.636 | 0.10 | 0.159 | 0.487 |
| Data set 3 |  |  |  |  |  |  |  |
| EGuNH | 307.586 | 621.173 | 626.240 | 1.401 | 0.226 | 0.151 | 0.317 |
| ENH | 310.775 | 627.566 | 632.616 | 1.912 | 0.319 | 0.199 | 0.084 |
| GaNH | 308.249 | 622.499 | 627.566 | 1.494 | 0.243 | 0.155 | 0.284 |
| LxNH | 313.300 | 632.601 | 637.668 | 2.176 | 0.369 | 0.172 | 0.183 |
| TLNH | 316.845 | 639.691 | 644.757 | 2.868 | 0.500 | 0.184 | 0.131 |
| MONH | 307.698 | 622.019 | 627.107 | 1.626 | 0.235 | 0.161 | 0.289 |
| NH | 400.885 | 805.770 | 809.184 | 2.518 | 0.433 | 0.607 | 0.000 |

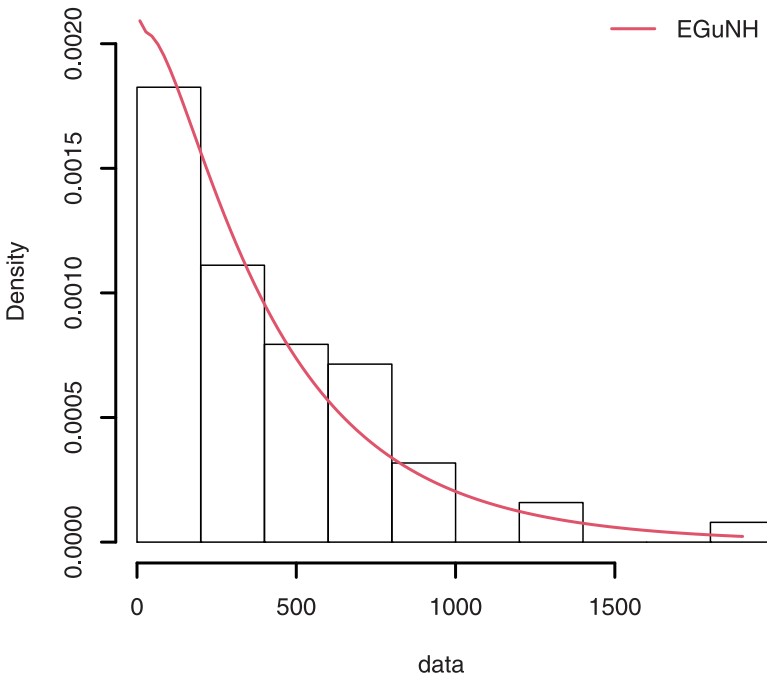

**Fig 23. Estimated density for D1.**

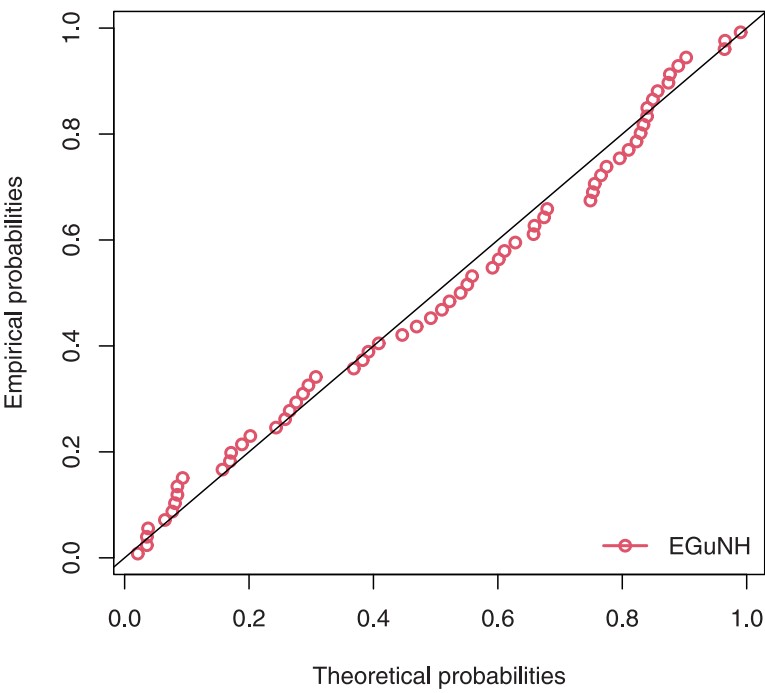

**Fig 24. Estimated plots of density for D2.**

**Empirical and theretical CDFs**

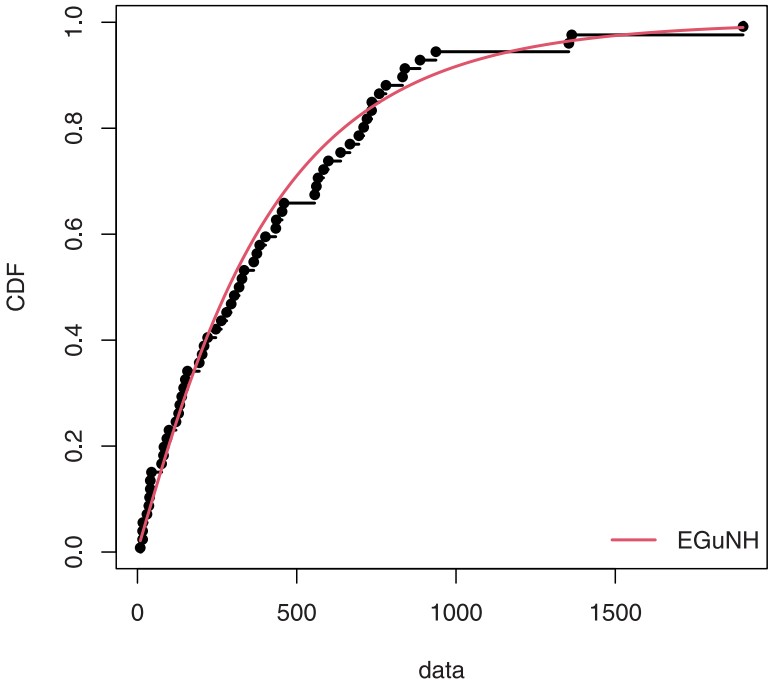

**Fig 25. Estimated density plot for D3.**

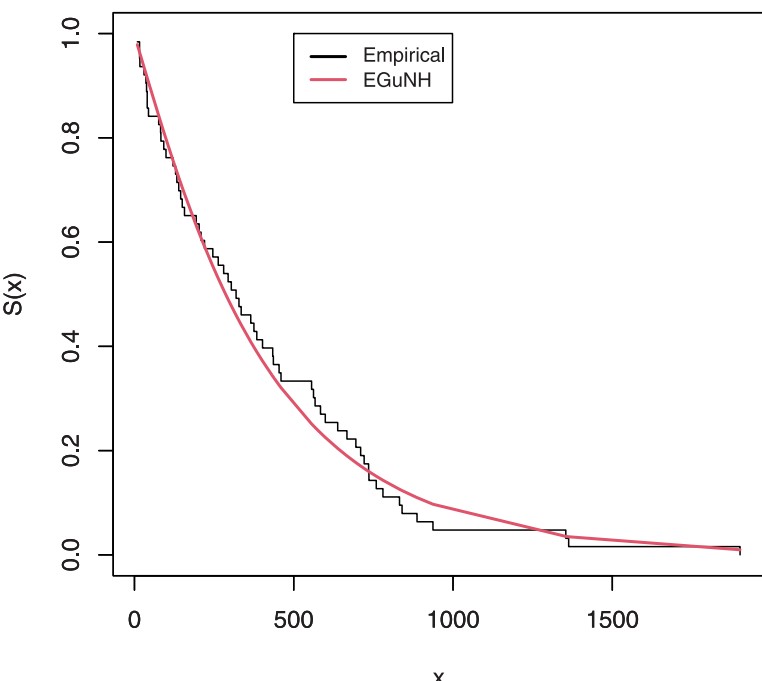

**Fig 26. PP plot for D1.**

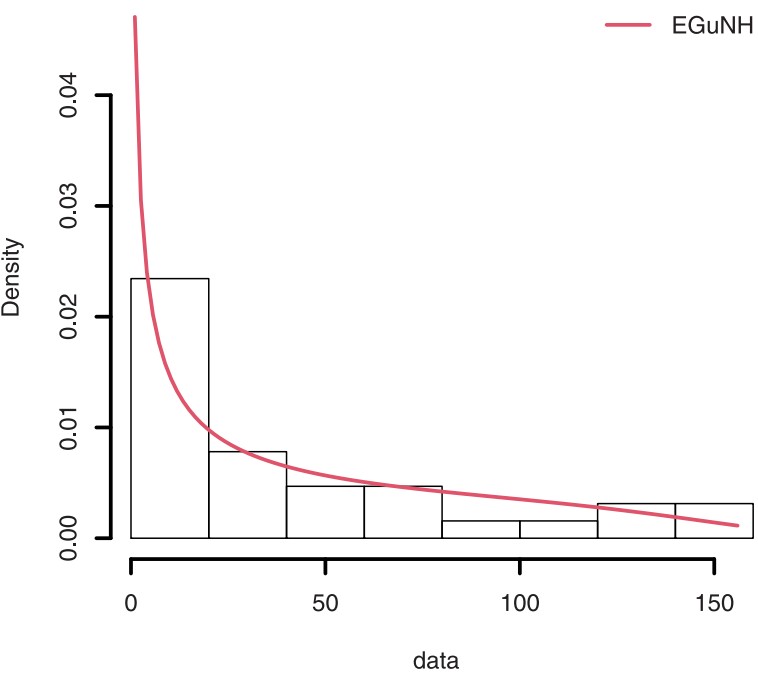

**Fig 27. PP plot for D2.**

estimated cdfs Figs (29)–(31) and estimated sfs Figs (32)–(34) for the three data sets are given. On the given data sets, the numerical values authenticates that the EGuNH model provides the best fit as compared to the other models.

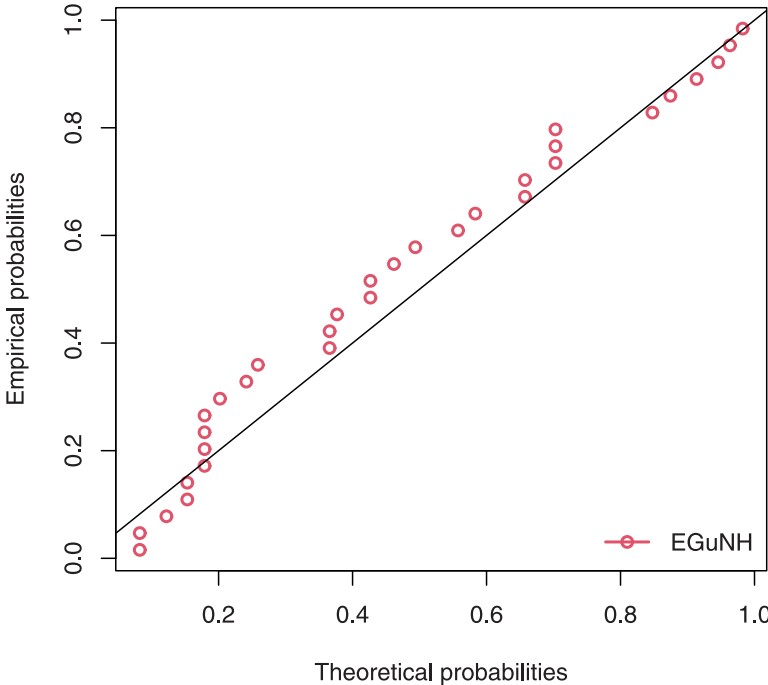

**Fig 28. PP plot for D3.**

**Empirical and theretical CDFs**

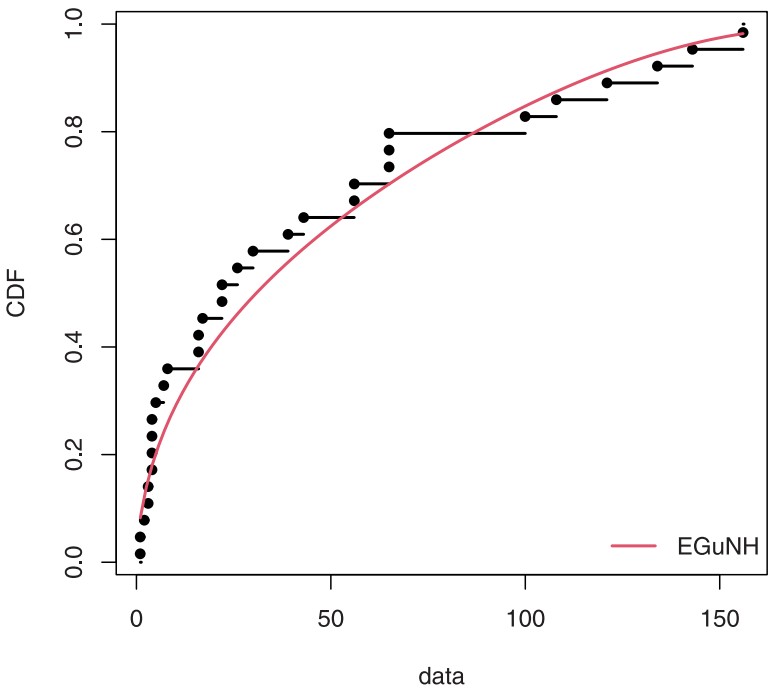

**Fig 29. Estimated cdf plot for D1.**

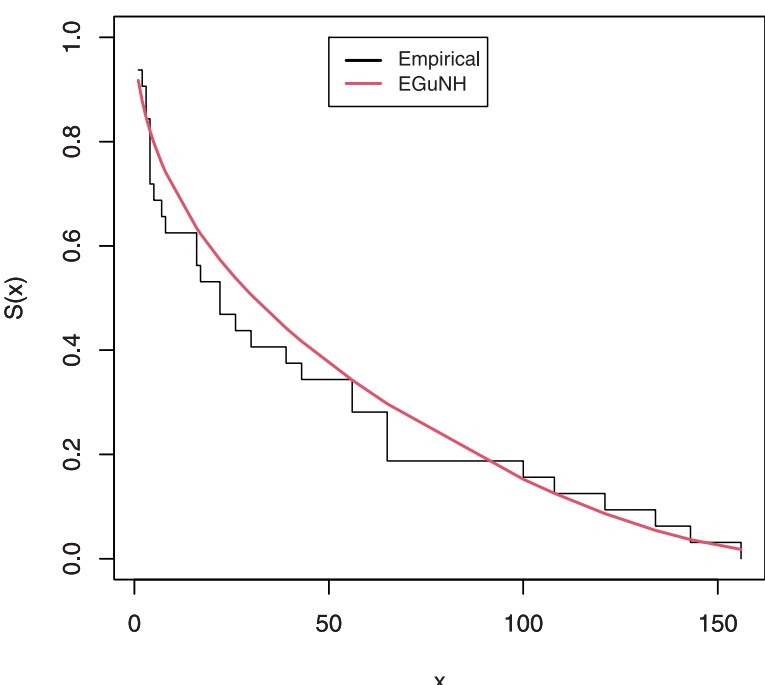

**Fig 30. Estimated cdf plot for D2.**

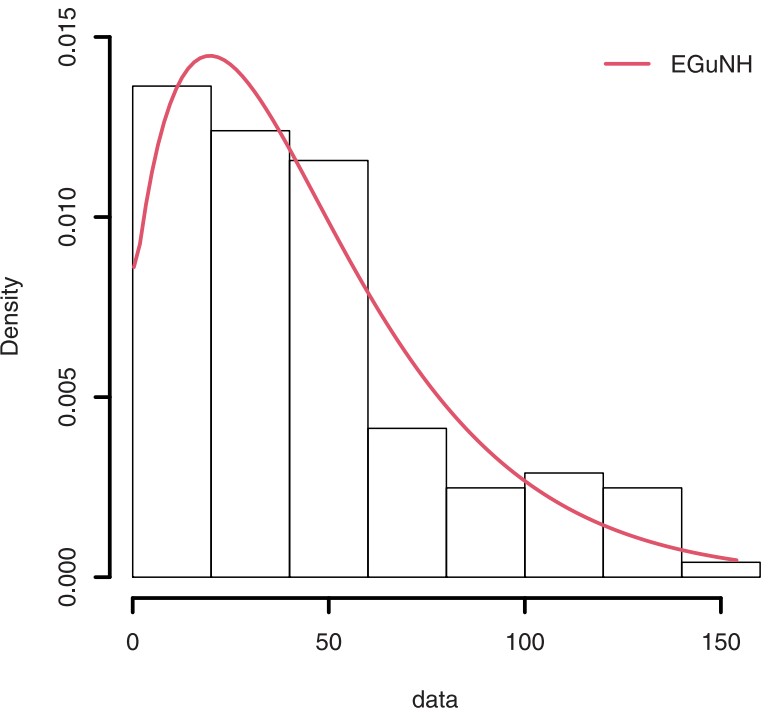

**Fig 31. Estimated cdf plot for D3.**

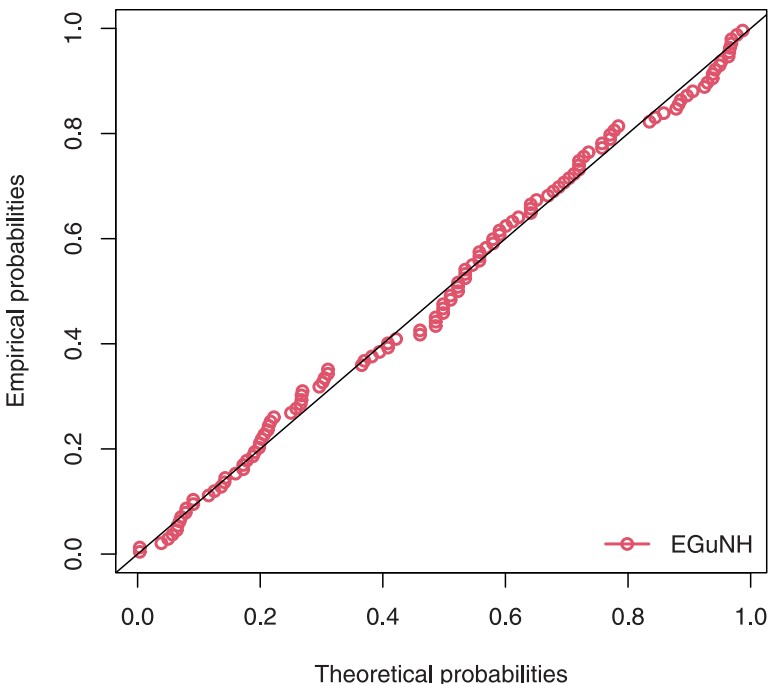

**Fig 32. Estimated sf plot for D1.**

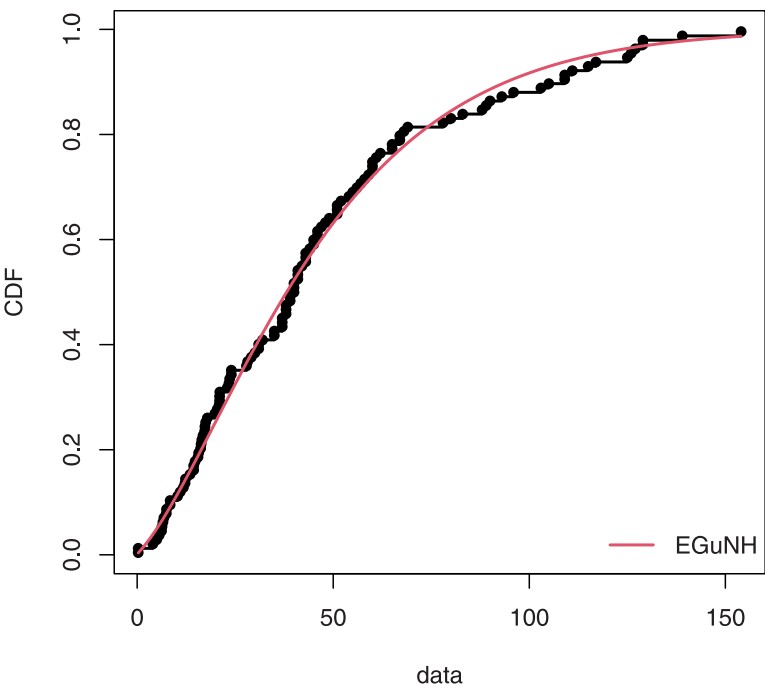

**Fig 33. Estimated sf plot of density for D2.**

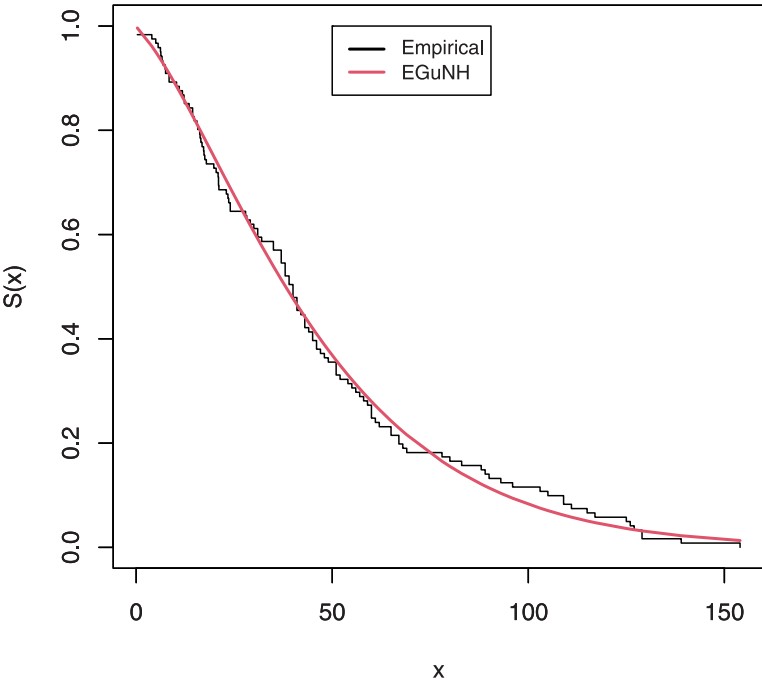

**Fig 34. Estimated sf plot for D3.**

**Table 12. Numerical measures of VaRs and ESs of EGuNH for D1, D2 and D3.**

| LoS | Data 1 | | Data 2 | | Data 3 | |
|---|---|---|---|---|---|---|
| | *VaRs* | *ESs* | *VaRs* | *ESs* | *VaRs* | *ESs* |
| 0.55 | 327.4119 | 146.4568 | 37.86083 | 12.29339 | 1125.354 | 727.9202 |
| 0.60 | 373.4939 | 163.4189 | 45.77802 | 14.74769 | 1201.044 | 764.1592 |
| 0.65 | 425.7544 | 181.5435 | 54.61479 | 17.46849 | 1282.559 | 800.8571 |
| 0.70 | 486.2076 | 201.0896 | 64.41790 | 20.46624 | 1372.164 | 838.4086 |
| 0.75 | 557.9849 | 222.4155 | 75.25685 | 23.75171 | 1473.216 | 877.2859 |
| 0.80 | 646.3563 | 246.0443 | 87.25890 | 27.33904 | 1591.174 | 918.1102 |
| 0.85 | 761.2607 | 272.8042 | 100.69662 | 31.25081 | 1736.442 | 961.7964 |
| 0.90 | 925.2607 | 304.1764 | 116.24233 | 35.52842 | 1927.397 | 1009.9029 |
| 0.95 | 1210.5273 | 343.4825 | 136.04737 | 40.26890 | 2210.153 | 1065.6140 |

The variance-covariance matrices of the MLEs of the EGuNH distribution for D1 is

$$\begin{pmatrix} 1.38801706 & 0.837248478 & 6.020134 \\ 0.83724848 & 0.004203568 & -3.061557 \\ 6.02013423 & -3.061556821 & 5.029424 \end{pmatrix}$$

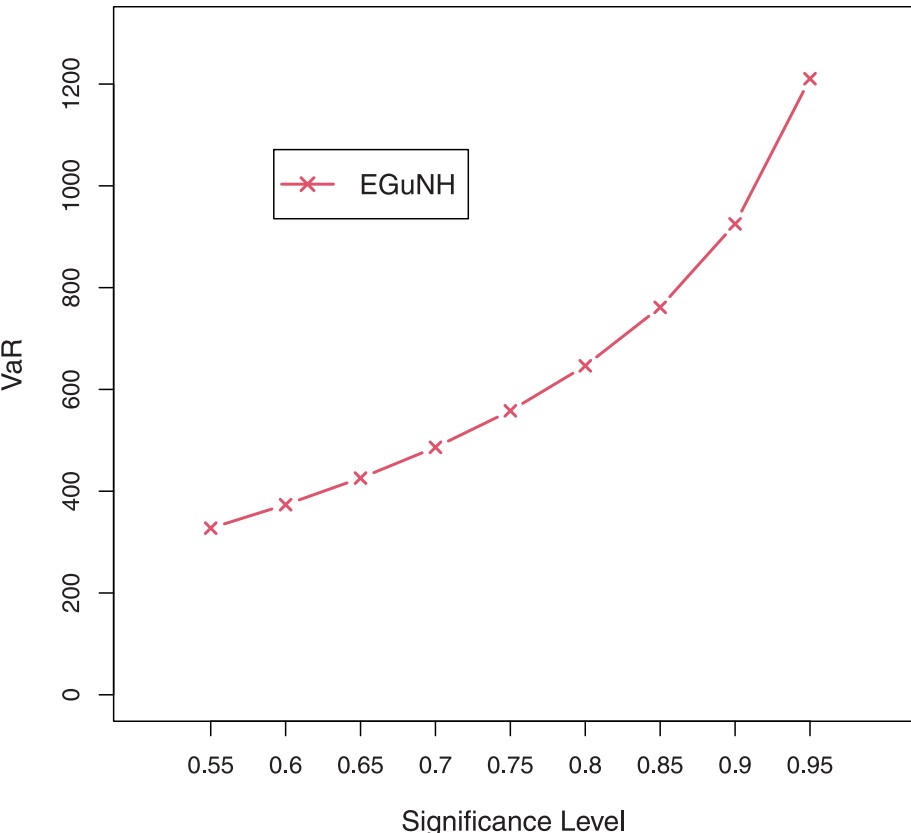

**Fig 35. Estimated VaRs for D1.**

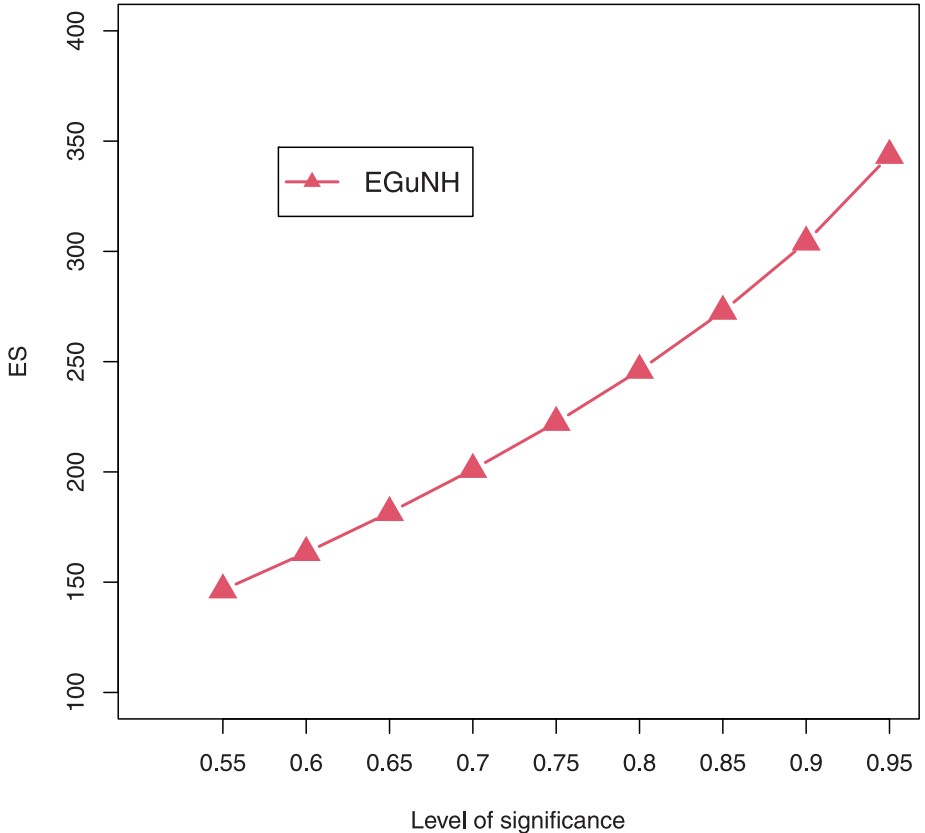

**Fig 36. Estimated VaRs for D2.**

The variance-covariance matrices of the MLEs of the EGuNH distribution for D2 is

$$\begin{pmatrix} 2.38801706 & -0.53724008 & 3.02013423 \\ -0.53724008 & 1.114203508 & 0.9611556827 \\ 3.02013423 & 0.911556827 & 11.55029424 \end{pmatrix}$$

The variance-covariance matrices of the MLEs of the EGuNH distribution for D3 is

$$\begin{pmatrix} 0.38801706 & -0.037248478 & 9.020134 \\ -0.03724848 & 0.004203568 & -5.061557 \\ 9.02013423 & -5.061556821 & 15.029424 \end{pmatrix}$$

**4.2.1 Numerical calculations of VaRs and ESs.** We were able to further investigate EGuNH's application to these risk measures thanks to the results reported in Section 4. To quantify the volatility associated with these measures, we take the values of MLEs of D1, D2 and D3., respectively, from Table 11. Higher risk measures indicate heavier tails, while lower risk measures indicate a model with a much lighter tail tendency. It's pertinent to mention that the EGuNH model yielded significantly more impressive results than others, implying that the model has a longer tail. The numerical findings of VaRs and ESs for data 1, data 2, and data 3

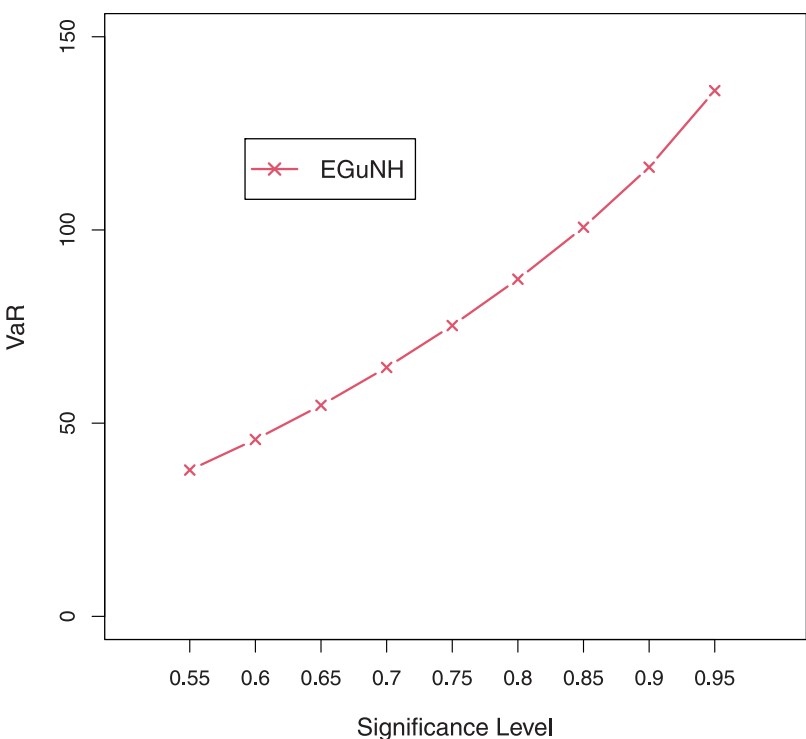

**Fig 37. Estimated VaRs for D3.**

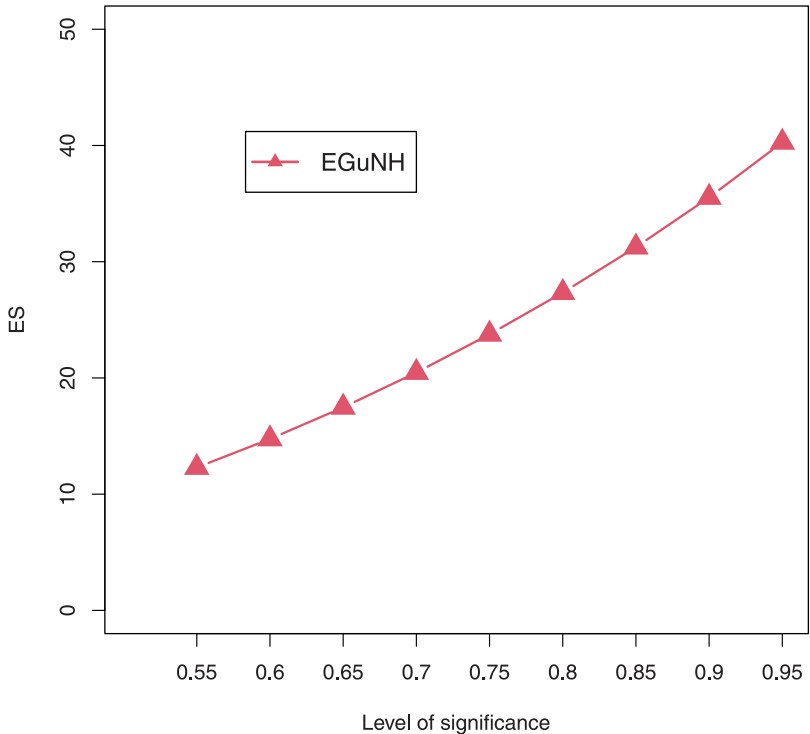

**Fig 38. Estimated ES for D1.**

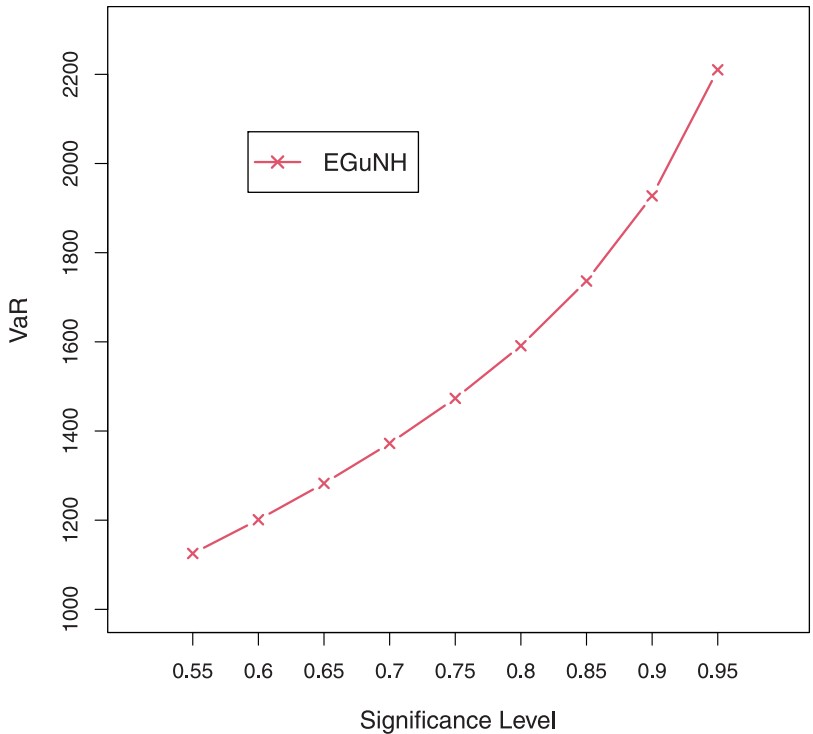

**Fig 39. Estimated ES for D2.**

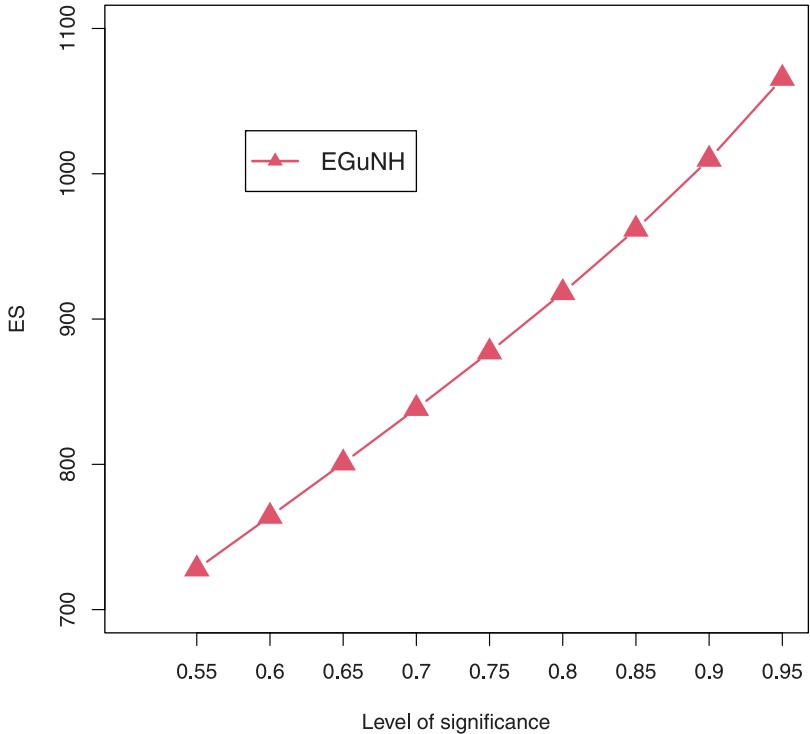

**Fig 40. Estimated ES for D3.**

of the proposed model at respective level of significance (LoS) are shown in Table 12. The summarized output of these risk measures (VaRs in Figs 35–37 and ESs in Figs 38–40), graphically, for the reader's expedience.

## 5 Concluding remarks

We propose and study the *EGuG* model and obtain a wide range of mathematical and statistical modelling methods to characterisethe model's structural and dynamic aspects including properties such as quantile function, ordinary and incomplete moments, mean deviations, bonferroni and lorenz curves, generating function and order statistics. The parameters of the family are estimated by the method of maximum likelihood. An extended exponential distribution is taken as baseline model to propose EGuNH distribution. Some simulations are performed to check the asymptotic properties of the estimates. Three applications to real data set are presented to illustrate the potentiality of the proposed models. For future research, the proposed model can further be extended using compounding. We expect that the modification may facilitate in estimating analytically tractable Bayesian estimates of the reliability function under different priors.

## Acknowledgments

The authors would seek this opportunity to thank the respected comments made by the reviewers which greatly help in the overall presentation of the manuscript.

## Author Contributions

**Conceptualization:** Aisha Fayomi.

**Data curation:** Sadaf Khan.

**Formal analysis:** Sadaf Khan.

**Investigation:** Sadaf Khan, Ali Algarni.

**Methodology:** Aisha Fayomi.

**Project administration:** Farrukh Jamal.

**Resources:** Muhammad Hussain Tahir.

**Supervision:** Muhammad Hussain Tahir.

**Validation:** Ali Algarni.

**Visualization:** Sadaf Khan.

**Writing – original draft:** Sadaf Khan.

**Writing – review & editing:** Farrukh Jamal, Reman Abu-Shanab.

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
