## [Decision Letter · Decision Letter 0]

9 Feb 2022

PONE-D-22-00530A New Extended Gumbel distribution: Properties

and ApplicationPLOS ONE

Dear Dr. Khan,

Thank you for submitting your manuscript to PLOS ONE. After careful consideration, we feel that it has merit but does not fully meet PLOS ONE’s publication criteria as it currently stands. Therefore, we invite you to submit a revised version of the manuscript that addresses the points raised during the review process.

The manuscript were reviewed by three experts, and their comments are included. Please carefully address all the comments, especially the comment on the originality of my paper.

We look forward to receiving your revised manuscript.

Kind regards,

Academic Editor

PLOS ONE

Journal Requirements:

2. Please include in your Methods section further details of how the dataset was obtained for the study, and provide the data sources.

3. Please ensure that you refer to Figures 4 and 6 in your text as, if accepted, production will need this reference to link the reader to the figure.

4. We note you have included a table to which you do not refer in the text of your manuscript. Please ensure that you refer to Table 7 in your text; if accepted, production will need this reference to link the reader to the Table.

Additional Editor Comments:

The manuscript were reviewed by three experts, and their comments are included. Please carefully address all the comments, especially the comment on the originality of my paper.

Reviewers' comments:

Reviewer's Responses to Questions

**Comments to the Author**

1. Is the manuscript technically sound, and do the data support the conclusions?

Reviewer #1: Yes

Reviewer #2: No

Reviewer #3: Yes

2. Has the statistical analysis been performed appropriately and rigorously? 

Reviewer #1: Yes

Reviewer #2: Yes

Reviewer #3: Yes

3. Have the authors made all data underlying the findings in their manuscript fully available?

Reviewer #1: Yes

Reviewer #2: Yes

Reviewer #3: Yes

4. Is the manuscript presented in an intelligible fashion and written in standard English?

Reviewer #1: Yes

Reviewer #2: No

Reviewer #3: Yes

5. Review Comments to the Author

Reviewer #1: This paper proposed a new generalization of the Gumbel distribution as well as three separate models evolved in the same family of distribution. As the author claimed, this should be the first time that the Nadarajah-Haghighi(NH) distribution is chosen as the baseline model to apply to the EGuG family. They formed the EGuNH distribution, conducted completed simulation study and applied it to lift data sets.

In all, this is a well-organized and written work, the idea and method are innovative and thoughtful. The reviewer would suggest the following modifications:

1. When claiming the start of limit of integers could be changed in formula (11), the author need to clarify why it could be verified by Mathematica, maybe list the result given by Mathematica, to make the point more convincing.

2. In section 3.6, there are 4 groups of parameter combinations but only 3 out of them are tested, the author should explain why they chose not to test all of them and explain how to decide which one to be tested.

3. The author should make some explanation or do some analysis for figure 5 instead of just leaving it there

4. At the beginning of section 3, the citation “The cdf and pdf…” should be Section 2.1.3 instead of 1.1.3

5. There are many typesetting errors and some punctuation errors, the author need carefully check the paper.

Reviewer #2: Thank you for the opportunity to review the manuscript. This work builds on prior work on Gumbel distribution. The authors studied the exponentiated Gumbel-G (EGuG) model and proposed a new model (called the EGuNH model) by extending the EGuG model. The new model takes the Nadarajah-Haghighi distribution as the baseline model. The simulations and real data are used to examine the model properties. The proofs in the paper seem to be mathematically correct. However, I have some concerns about the paper I would like to discuss with the authors.

Major concerns relating to methods or significance:

This work is very similar to A. A. Ogunde et.al (https://www.hindawi.com/journals/jam/2020/2798327/ ) and Hormatollah Pourreza et.al (https://journals.sagepub.com/doi/full/10.1177/09622802211009262 ). Did authors read these papers to compare the result? In addition, the manuscript is not clearly presented overall. It seems to me that this paper did not add much to the scientific value in terms of combined level of methodological and practical innovation.

Minor concerns related to clarity:

1. In the Abstract, the authors used the term “greatest likelihood method”. Is it the maximum likelihood method? If so, I think the common term should be the maximum likelihood method.

2. In Section 1, the authors did not explain what T-X methodology is. It would be good if the authors can provide a citation.

3. In Section 2, it would be good to use the hazard rate function to define “hrf” rather than failure rate function.

4. In-Page 13, the authors did not cite figure # for “Plots of MGs for some parameter values”.

5. The tables should be self-explanatory. Please explain the acronyms in the tables. For example, what is LoS in Table 12?

6. The authors seem did not explain most of figures and tables. For example, what is the message for Figure 13?

7. In the simulations, it would be good if the authors could report the coverage probability of the 95% confidence intervals as another evaluation metric for MLE estimation of model parameters.

8. In real data examples, the authors did not show how to interpret model parameters and their practical meanings in real data.

Reviewer #3: In the manuscript “A New Extended Gumbel distribution: Properties and Application”, Fayomi et al. introduced a new generalization of the Gumbel distribution. The author presented the distribution through the data simulation as well as three real-world applications.

The paper is very well presented, and the author provided a good review of the literature. The equations are well explained and clearly presented; I am glad to see that the author used the real-world applications to show the superiority of the distribution.

I would have like to see how the extended Gumbel distribution behaves for global sequence alignment application. For example, Sardiu et al. 2005, showed that the score statistics of sequence alignment follows a Gumbel distribution. Is the extended Gumbel distribution presented in this paper superior to the classical Tracy-Widom distribution for example for global alignments? I suggest the author cite this paper since the application of the Gumbel distribution in biological applications were used since 2005. The author does not talk in the manuscript about the Tracy-Widom distributions. This needs to be included in the introduction.

I would also suggest that the author comment more on the limitation of this distribution. How the distribution change when the size is large or too small, for example. Is it a critical point where the distribution does not apply anymore? These are important points which need to be addressed to prove superiority and utility of the extended Gumbel distribution.

Overall, the manuscript is good, however I suggest that the author address these concerns.

6. PLOS authors have the option to publish the peer review history of their article (what does this mean?). If published, this will include your full peer review and any attached files.

Reviewer #1: **Yes**

Reviewer #2: No

Reviewer #3: No

---

## [Author Response · Author response to Decision Letter 0]

12 Mar 2022

It was the evident wish and will of the Reviewers that we make the changes to our manuscript on their insightful observations. Without losing the originality, we have tried to accommodate their suggestions in best possible manner. We have also improved the overall presentation of the paper. 

We thank the Editor and the three referees for the constructive comments and hope that the revision is now appropriate for publishing.

---

## [Decision Letter · Decision Letter 1]

4 Apr 2022

A New Extended Gumbel distribution: Properties

and Application

PONE-D-22-00530R1

Dear Dr. Khan,

We’re pleased to inform you that your manuscript has been judged scientifically suitable for publication and will be formally accepted for publication once it meets all outstanding technical requirements.

Kind regards,

Academic Editor

PLOS ONE

Additional Editor Comments (optional):

The revised manuscript was reviewed by two experts. Both of them believe that all comments have been addressed. I agree with them and suggest acceptance.

Reviewers' comments:

Reviewer's Responses to Questions

**Comments to the Author**

1. If the authors have adequately addressed your comments raised in a previous round of review and you feel that this manuscript is now acceptable for publication, you may indicate that here to bypass the “Comments to the Author” section, enter your conflict of interest statement in the “Confidential to Editor” section, and submit your "Accept" recommendation.

Reviewer #1: All comments have been addressed

Reviewer #3: All comments have been addressed

2. Is the manuscript technically sound, and do the data support the conclusions?

Reviewer #1: Yes

Reviewer #3: Yes

3. Has the statistical analysis been performed appropriately and rigorously? 

Reviewer #1: Yes

Reviewer #3: Yes

4. Have the authors made all data underlying the findings in their manuscript fully available?

Reviewer #1: Yes

Reviewer #3: Yes

5. Is the manuscript presented in an intelligible fashion and written in standard English?

Reviewer #1: Yes

Reviewer #3: Yes

6. Review Comments to the Author

Reviewer #1: (No Response)

Reviewer #3: All concerns have been addressed. The author addressed my questions. No need for further work. I suggest this paper for acceptance.

7. PLOS authors have the option to publish the peer review history of their article (what does this mean?). If published, this will include your full peer review and any attached files.

Reviewer #1: No

Reviewer #3: No

---

## [Editor Report · Acceptance letter]

18 Apr 2022

PONE-D-22-00530R1 

A New Extended Gumbel Distribution: Properties And Application 

Dear Dr. Khan:

I'm pleased to inform you that your manuscript has been deemed suitable for publication in PLOS ONE. Congratulations! Your manuscript is now with our production department. 

Kind regards, 

on behalf of

Dr. Jiangtao Gou 

Academic Editor

PLOS ONE